# Don't Take Things Out of Context: Attention Intervention for Enhancing Chain-of-Thought Reasoning in Large Language Models

**Shaotian Yan**[1], **Chen Shen**[1*], **Wenxiao Wang**[2,1], **Liang Xie**[3,1], **Junjie Liu**[1], **Jieping Ye**[1]

[1] Alibaba Cloud Computing
[2] College of Software, Zhejiang University
[3] Zhejiang University of Technology, College of Computer Science and Technology
`yanshaotian@gmail.com, zjushenchen@gmail.com`

## Abstract

Few-shot Chain-of-Thought (CoT) significantly enhances the reasoning capabilities of large language models (LLMs), functioning as a whole to guide these models in generating reasoning steps toward final answers. However, we observe that isolated segments, words, or tokens within CoT demonstrations can unexpectedly disrupt the generation process of LLMs. The model may overly concentrate on certain local information present in the demonstration, introducing irrelevant noise into the reasoning process and potentially leading to incorrect answers. In this paper, we investigate the underlying mechanism of CoT through dynamically tracing and manipulating the inner workings of LLMs at each output step, which demonstrates that tokens exhibiting specific attention characteristics are more likely to induce the model to take things out of context; these tokens directly attend to the hidden states tied with prediction, without substantial integration of non-local information. Building upon these insights, we propose a Few-shot Attention Intervention method (FAI) that dynamically analyzes the attention patterns of demonstrations to accurately identify these tokens and subsequently make targeted adjustments to the attention weights to effectively suppress their distracting effect on LLMs. Comprehensive experiments across multiple benchmarks demonstrate consistent improvements over baseline methods, with a remarkable 5.91% improvement on the AQuA dataset, further highlighting the effectiveness of FAI.

## 1 Introduction

Large Language Models (LLMs) have achieved significant advancements in tackling complex reasoning tasks (Zhou et al., 2023; Yao et al., 2023; Besta et al., 2023), such as mathematics(Imani et al., 2023; Cobbe et al., 2021; Yuan et al., 2023) and symbolic logic(Patel et al., 2021; Srivastava et al., 2022; Ling et al., 2017), by adopting the innovative Chain-of-Thought (CoT) prompting strategy (Wei et al., 2022) which promotes the LLMs' propensity to break down the thought process into multiple intermediary steps leading to the final answer.

The most prevalent paradigm of CoT is known as few-shot CoT, which comprises a handful of demonstrations, each consisting of a query paired with a reasoning chain. However, in practice, the performance of LLMs is sensitive to the selection of CoT demonstrations (Huang et al., 2023; Rubin et al., 2021; Luo et al., 2023; Liu et al., 2023; Su et al., 2022). Employing diverse CoT exemplars can cause considerable variations in the overall precision of LLMs. We further demonstrate that even when overall accuracy rates are comparable, varying CoT demonstrations can lead to substantial differences in the distribution of specific questions that are answered correctly versus those answered incorrectly. This inconsistency raises concerns about the robustness of CoT and presents a crucial

---

*Corresponding author

challenge for its real-world application. Yet the underlying cause of the observed performance variations remains largely unclear.

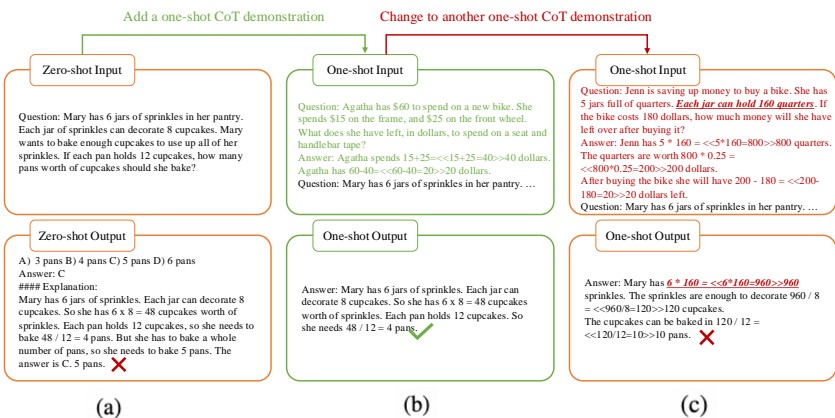

Figure 1: An example of a Few-shot Chain-of-Thought demonstration distracting the reasoning of LLMs. Questions are collected from the GSM8K (Cobbe et al., 2021) dataset, and the responses are generated by Llama-3-8B-Instruct (AI@Meta, 2024).

Recent studies (Madaan & Yazdanbakhsh, 2022; Tang et al., 2023; Wang et al., 2023a; Jin et al., 2024; Ye et al., 2023; Fu et al., 2022; Prystawski et al., 2024) have made efforts on identifying the factors that influence the effectiveness of CoT demonstrations. (Wang et al., 2023a) shows that employing invalid reasoning steps within the CoT demonstrations can still yield performance that is nearly comparable to using valid reasoning. They further emphasize that the performance relies more critically on the logical consistency of the rationale and its relevance to the corresponding question. (Jin et al., 2024) finds that increasing reasoning steps in the CoT demonstrations can improve the overall accuracy of LLMs across multiple datasets via contrastive experiments. However, the insights presented by these existing studies are primarily based on the overall accuracy across the entire dataset, failing to illuminate the instance-specific fluctuations inherent of CoT.

While existing studies primarily focus on the positive effects of CoT, in this paper, motivated by the aforementioned fragilities of CoT, we find that few-shot CoT does not always perform in the way we expect. Specifically, in addition to the commonly known positive instructional effect of CoT demonstrations, which encourages LLMs to output intermediate reasoning steps leading to the final answer, certain localized information within the demonstrations can unexpectedly distract the reasoning process of LLMs in an undesirable manner. As shown in Figure 1, the model inaccurately incorporates information from demonstrations (i.e., "each jar can hold 160 quarters" which is bolded and underlined in Figure 1(c).) while generating intermediate reasoning steps in its output process, ultimately leading to wrong answers. In other cases, despite LLMs not directly copying information from demonstrations, their outputs are nonetheless implicitly distracted by specific pieces of information or tokens within the demonstrations.

Ideally, LLMs should treat CoT demonstrations as a whole, learning and mimicking the implied style and higher-order semantics within them. They should apply this knowledge flexibly, rather than simply taking the content of the demonstrations out of context, memorizing it, or being distracted by it. Therefore, the distracting effect should be alleviated to enhance the robustness of CoT reasoning. Nevertheless, it is closely intertwined with the complete semantics of the demonstration. Efforts to simply remove some tokens may disrupt the overall meaning of the demonstration, thereby diminishing the effectiveness of the CoT demonstration. This dilemma prompts a deeper examination of the internal workings of LLMs.

By visualizing the interactions, quantified through commonly used attention saliency scores, between tokens at each layer and time step, we observe that some tokens in the demonstrations have

the following attention characteristics: these tokens do not significantly gather information from other tokens (meaning that the hidden state corresponding to these tokens retains a considerable amount of their own semantic information) but can directly channel towards the prediction position at certain time steps. The behavior of the cases depicted in Figure 1(c) aligns closely with the information flow characteristics exhibited by these tokens, and indeed, we observe such features in the case presented in Figure 1 as well as in a quantitative experiment which involves about 180 other cases where the model is disrupted (see section 2 for detail). This mirrors human cognitive tendencies, where our attention is often disproportionately drawn towards salient local elements, leading to an inadvertent overemphasis on these aspects at the expense of more fundamental global context. Given their retention of substantial semantic integrity, these tokens are particularly susceptible to becoming prominent focal points under certain conditions, thereby influencing information processing dynamics.

To address the aforementioned issue, we introduce a **F**ew-shot **A**ttention **I**ntervention (FAI) technique that dynamically analyzes the attention patterns of demonstrations to accurately identify tokens with rather isolated semantics. By making targeted adjustments to the attention weights, FAI can block the information flow from these tokens to the output token, effectively suppressing their distracting effect on LLMs. Comprehensive experimentation across various reasoning benchmarks demonstrates that with only lightweight and efficient interventions—comprising about 15% of the tokens in the GSM8K demonstration—FAI can consistently enhance the performance of LLMs in few-shot Chain of Thought (CoT) scenarios. Notably, the implementation of FAI leads to a remarkable 5.91% improvement on the AQuA (Ling et al., 2017) dataset.

## 2   CASE ANALYSIS OF INFORMATION FLOW INSIDE LLMS WITH FEW-SHOT COT

Suppressing the distracting effect within input text is challenging, prompting us to delve deeper into the internal mechanisms of large language models. Saliency techniques (Simonyan et al., 2013) are commonly utilized for analyzing the flow of information within a model. By comprehensively considering attention scores and gradients, the saliency score can measure the significance of information interaction between tokens. Existing works often calculate the saliency score based on the model's output at either the answer step or the last step (Li et al., 2024; Wang et al., 2023b), yet we argue that this single-step approach may overlook crucial information in reasoning tasks that employ Chain-of-Thought demonstrations. This is because the CoT demonstration does not necessarily have a direct effect on the final output of the model, but rather, it influences the answer indirectly by impacting the way the model generates its rationale. Therefore, we propose to dynamically trace the inner workings of LLMs and visualize the attention interactive pattern at each output step.

Following common practice (Wang et al., 2023b; Michel et al., 2019), we leverage the Hadamard product of attention weight and its corresponding gradient matrix towards the loss $\mathcal{L}$ of output token to calculate the saliency score matrix $S_{l,t}$ for layer $l$ at $t$-th output step:

$$S_{l,t} = \left| \sum_h (A_{h,l,t}) \odot \frac{\partial \mathcal{L}(x_t)}{\partial A_{h,l,t}} \right| \tag{1}$$

where $A_{h,l,t}$ is the attention weight for the $h$-th attention head in layer $l$ at $t$-th output step and $x_t$ is the output token for $t$-th time step. Consequently, $S_{l,t}(i,j)$ can measure the significance of the information flow from the $j$-th token to $i$-th token in layer $l$ of time step $t$.

Figure 2 presents four samples along with their corresponding attention salience analyses. For each sample, the upper part of the figure shows the salience scores of the demonstration tokens, the tokens in the question, and the generated tokens for the current prediction position; darker colors indicate stronger salience. Subsequently, we select one token that has a significant impact on the output, and the lower part of the figure displays the salience of the preceding tokens for that selected token. In each case, we choose a layer with pronounced phenomena to demonstrate the characteristics of attention salience more intuitively.

Figure 2 (a) and (c) illustrate two examples in which the token has already encoded relatively global information from other tokens in previous layers, significantly influencing the model's output without leading to incorrect answers. The example in Figure 2 (b) demonstrates how the token "160"

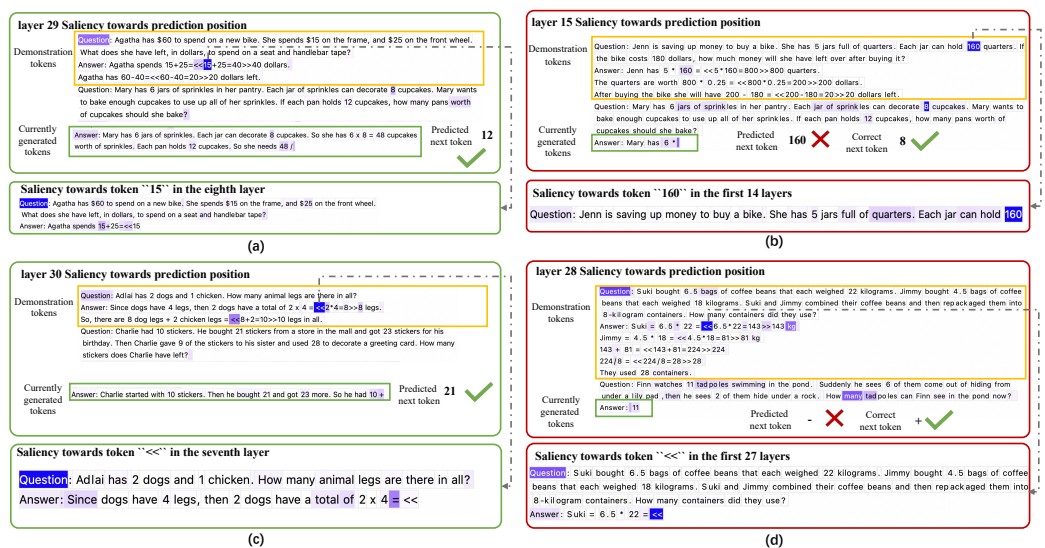

Figure 2: Attention salience analysis example. For each sample, the upper part of the figure shows the salience scores of the demonstration tokens, the tokens in the question, and the generated tokens for the current prediction position; darker colors indicate stronger salience. Subsequently, we select one token that has a significant impact on the output, and the lower part of the figure displays the salience of the preceding tokens for that selected token. In each case, we choose a layer with pronounced phenomena to demonstrate the characteristics of attention salience more intuitively. (a)(c): Cases correctly answered. (b)(d): Cases with wrong responses.

in the demonstration has a profound impact on the model's output, disrupting its reasoning process. Prior to this, no other tokens in the preceding layers exhibit notable information convergence toward "160." (d) presents another similar case; however, unlike Figure 2 (b), it is not immediately clear from a semantic perspective what influence the tokens in the demonstration have on the model's output. Nevertheless, this case shares similar information flow characteristics with Figure 2 (b).

To further understand the relationship between these information flow characteristics and the model's tendency to misinterpret contexts, we conducted multiple few-shot experiments on GSM8K to construct a dataset of samples where model responses were influenced by demonstrations, leading to errors (the construction method is detailed in section 4.2). We randomly sampled 180 of these samples for manual observation. The types of errors can be broadly categorized into four categories: (i) **I**ncorporating information from **F**ew-shot examples (**IF**), (ii) **M**athematical **C**alculation errors (**MC**), (iii) errors in **R**easoning **S**teps (**RS**), and (iv) errors from **R**epeated **O**utputs (**RO**).

The distribution of these errors is presented in the table below.

Table 1: Error case Analysis.

| Error Types | IF | MC | RS | RO |
|---|---|---|---|---|
| Number of Samples | 17 | 41 | 57 | 65 |

We analyze these samples using the attention saliency method described in this section and find that almost all the IF samples, as well as most of the MC and RS samples, result in erroneous outputs due to the distracting effect. However, the majority of the RO samples do not have errors caused by the distracting effect. Based on this, it is estimated that about 60% of the erroneous responses are due to the distracting effect. Details about these samples can be found in section A.2 in Appendix.

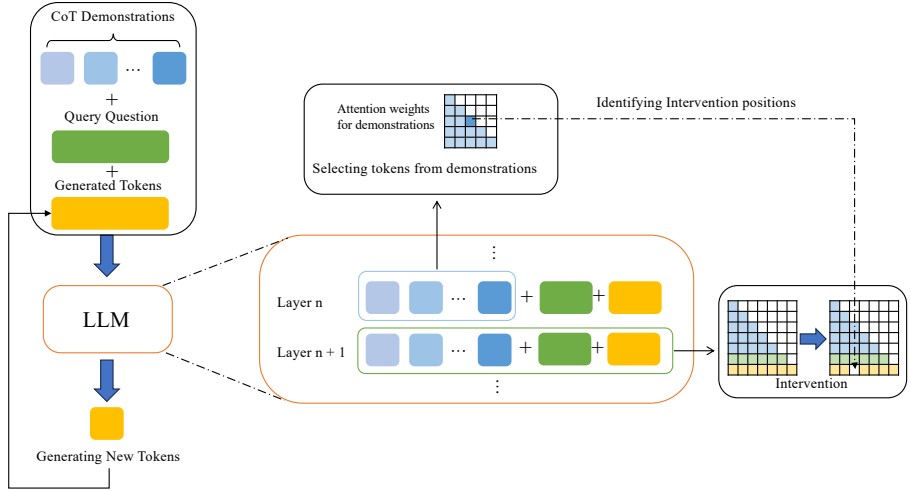

Figure 3: Overview of the proposed FAI method. In each layer, FAI identifies the positions that require intervention by analyzing the attention matrix from the demonstration. It then applies these interventions to the attention matrix of the subsequent layer, relative to the positions of the output tokens.

# 3 METHOD

## 3.1 OVERVIEW

This section describes the Few-shot Attention Intervention method (FAI) we propose in detail. As shown in Figure 3, in each layer, FAI identifies the positions that require intervention by analyzing the attention matrix from the demonstration. It then applies these interventions to the attention matrix of the subsequent layer, relative to the positions of the output tokens. Since the attention matrix is inherently computed by LLMs, FAI, apart from analyzing the positions requiring intervention (which constitutes a very lightweight computation), introduces virtually no additional computational overhead. This renders FAI of low complexity and high efficiency.

## 3.2 IDENTIFYING POSITIONS REQUIRING INTERVENTION

As discussed above, a key characteristic of tokens that can lead the model to take things out of context is that they influence the output token before significant information from other tokens has been aggregated. Therefore, an index is needed to measure the intensity of information aggregation to identify certain tokens. While the saliency score emerges as a potent candidate for this task, the high computational overhead associated with calculating saliency at every step through back propagation necessitates an alternative. In view of that, we instead leverage the attention score to construct an aggregation coefficient $\alpha$ to measure how much information is aggregated. Specifically, given a LLM with $L$ layers and each layer consists of $H$ attention heads, for every token $t_i$ in demonstrations, we first calculate its average attention score across all the heads in layer $l$:

$$\overline{\mathcal{A}}_l = \frac{\sum_{h=0}^{H} \mathcal{A}_l^h}{H} \tag{2}$$

where $\mathcal{A}_l^h$ refers to the attention weight matrix at the $h$-th attention head of layer $l$. We then define the value of $\overline{\mathcal{A}}_l(t_i, t_i)$ as the aggregation coefficient $\alpha$. For token $t_i$ at layer $l$:

$$\alpha_l^{t_i} = \overline{\mathcal{A}}_l(t_i, t_i) \tag{3}$$

where $\overline{\mathcal{A}}_l(t_i, t_i)$ refers to the attention score of token $t_i$ to itself at layer $l$. The underlying reason behind this formulation is that, since the attention matrix $\mathcal{A}_l^h$ has already been normalized, a low

attention score $\overline{\mathcal{A}}_l(i, i)$ for a particular position $i$ implies that other tokens have higher attention scores for the current position (i.e., where information aggregating is likely to happen) while a higher attention score suggests less aggregation.

Next, we define a threshold $\tau$ to determine whether the information aggregation of a token is significant.

$$\tau = \frac{\lambda}{index_{t_i}} \tag{4}$$

where $index_{t_i}$ refers to the index of token $t_i$ in the demonstration it belongs to. Given that the attention matrix $\mathcal{A}_l^h$ has already been normalized, the term $\frac{1}{index_{t_i}}$ is approximately equal to the mean of the attention scores directed towards token $t_i$ provided that the attention scores are uniformly distributed within the same demonstration. Here, $\lambda$ serves as a hyper parameter.

Therefore, we can consider that token $t_i$ has not experienced significant information aggregation at layer $l$ if $\alpha_l^{t_i}$ is larger than $\tau$.

### 3.3 Intervening Information Flow

In each layer $l$, we block the information flow of tokens that have not undergone significant aggregation up to the current layer toward the model's output token, which is done by setting the attention score from token $t_i$ towards the output token to zero in all the attention heads at layer $l$.

Note that the first token in the input prompt will never be blocked as it serves as an attention sink (Xiao et al., 2023a).

## 4 Experimental Results

### 4.1 Experiments on various reasoning tasks

Firstly, we validate the proposed FAI across a diverse range of datasets, encompassing various categories of reasoning tasks. For math reasoning, the test sets of the popular **GSM8K** (Cobbe et al., 2021) and **AQuA** (Ling et al., 2017) benchmarks are adopted, which contain 1319 and 254 real world math problems respectively. The prevalent Commonsense Question Answering (**CSQA**) (Talmor et al., 2018) dataset poses questions that delve into the realm of commonsense knowledge about the world, often requiring an understanding of intricate semantics and drawing on prior information. Complementing this, two challenging evaluation sets from Big-Bench-Hard (Srivastava et al., 2022) are utilized for comprehensive assessment. The **Date Understanding** task necessitates that language models extract a precise date from contextual clues embedded within the text, while the **Sport Understanding** task involves discerning the plausibility or implausibility of statements related to athletic events. The frequently used **Last letter concatenation** (Wei et al., 2022) task is leveraged to examine the performance of FAI in symbolic reasoning. Experiments are conducted using GPT2-XL, GPT-NEO(Radford et al., 2019), Llama-3-8B-Instruct and Llama-3-70B-Instruct(AI@Meta, 2024) with the coefficient $\lambda$ consistently set at 1. For all the datasets, we select four demonstrations from the paper of CoT (Wei et al., 2022). Other implementation details are provided in the Appendix. The experimental results are illustrated in Table 2. By implementing FAI, the model

Table 2: Overall Accuracy on various benchmarks

| Method | AQuA | GSM8K | CSQA | Date | Sport | Last letter |
|---|---|---|---|---|---|---|
| GPT2-XL | 22.44 | 2.27 | 16.54 | 2.0 | 55.2 | 0.0 |
| GPT2-XL + FAI | **28.74** | **2.88** | **16.63** | 2.0 | 55.2 | 0.0 |
| GPT-NEO | 22.83 | 1.59 | 22.69 | 3.2 | 54.4 | 0.0 |
| GPT-NEO + FAI | **36.22** | **2.50** | **23.26** | **3.6** | **55.2** | 0.0 |
| Llama3 8B | 40.94 | 70.32 | 71.17 | 64.00 | 95.60 | 58.67 |
| Llama3 8B + FAI | **46.85** | **71.24** | **74.28** | **65.60** | **96.00** | **62.00** |
| Llama3-70B | 66.14 | 91.28 | 77.31 | 87.60 | 97.2 | 84.00 |
| Llama3-70B +FAI | **66.53** | 91.28 | **78.62** | **88.0** | **98.0** | **85.33** |

has seen significant enhancements in overall accuracy across various tasks, notably achieving an impressive 5.91% improvement on the AQuA dataset.

## 4.2 ABLATION STUDIES

**Construction of Test Dataset.** To assess the effectiveness in alleviating the distracting effect of CoT on a normal dataset can be challenging, due to the various confounding factors contained in the dataset, such as some questions being answered incorrectly due to being distracted by CoT, while others are coincidentally answered correctly due to the influence of some tokens in the demonstration. To this end, we conduct a series of manipulations on the test set of GSM8K to construct two validation sets (i.e. $GSM_{good}$ and $GSM_{bad}$).

Firstly, we randomly sample 45 groups of CoT demonstrations from the train set of GSM8K and then analyze the one-shot performance of Llama-3-8B-Instruct on the test set of GSM8K. We find that there is considerable variability in overall accuracy across different demonstrations, with a range of 4.02 while the average accuracy is 67.7.

Table 3 presents the distribution of accuracy for each test sample across the 45 trials. As indicated in Table 3, only 198 out of the 1319 samples received consistent responses—either always correct or always incorrect—across the various demonstrations. The remaining samples, which make up approximately 85% of the test set, yielded varied outcomes depending on the demonstration, suggesting that the potential for success with few-shot chain-of-thought reasoning is substantial, yet the risk of failure is equally significant.

Here we employ the 146 samples in Table 3 that consistently get correct answers across various demonstrations as the $GSM_{good}$ set, since these samples are less likely to be influenced by the distracting effect of CoT. Meanwhile, the 347 samples with an accuracy rate higher than 90%, concatenated with the demonstration that led to incorrect answers, form the $GSM_{bad}$ set, because the incorrectness of these samples is more likely to be caused by the distracting effect. A detailed analysis of the samples in $GSM_{bad}$ can be found in section A.2 in Appendix.

**Compared Methods.** We employ a contrasting setting in which all the attention scores from the demonstration to the prediction position are set to zero at each layer, to further validate the correlation between the distracting effect of CoT and the aforementioned information flow phenomenon. All the experiments are conducted on Llama-3-8B-Instruct. RAFR( Rate of Answer Following the Rationale) stands for the ratio of LLMs generating rationales before the final answer. which can be used as an indicator of the degree to which the postive effect of CoT is preserved.

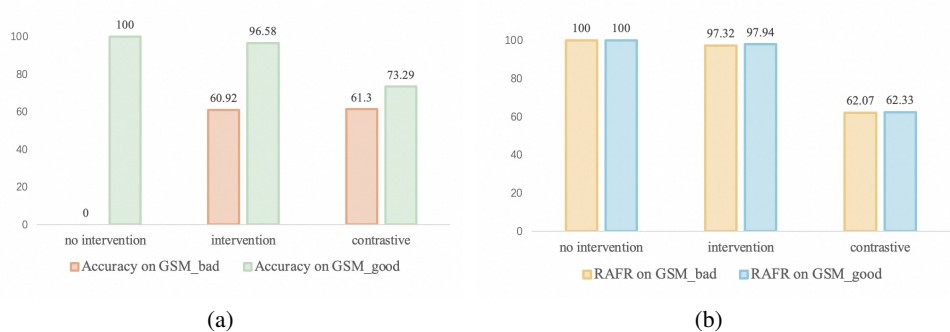

(a)             (b)

Figure 4: (a) Comparison of the Overall Accuracy on $GSM_{bad}$ and $GSM_{good}$. (b) Comparison of the Rate of Answer Following Rationale on $GSM_{bad}$ and $GSM_{good}$.

**Results and Analysis.** By locating and intervening with tokens that have not significantly converged, there has been a notable improvement in the $GSM_{bad}$ test set, demonstrating an association between this pattern of information transmission and the distracting effect of chain of thought. The contrastive setting, in which all information flow from demonstrations to prediction is blocked in each attention head, indeed, significantly inhibits the distracting effect of CoT, but it also leads to a substantial decrease in accuracy on $GSM_{good}$. Moreover, the RAFR metric significantly decreases

Table 3: Distribution of Accuracy for each test sample on GSM8K across 45 tries

| Accuracy | 100 | $> 90$ | $> 80$ | $> 70$ | $> 60$ | $> 50$ | $> 40$ | $> 30$ | $> 20$ | $> 10$ | $> 0$ | 0 |
|----------|-----|--------|--------|--------|--------|--------|--------|--------|--------|--------|-------|-----|
| Count | 146 | 347 | 166 | 140 | 67 | 81 | 64 | 64 | 48 | 69 | 75 | 52 |

on both datasets, indicating that while it suppresses the distracting effect, it also inhibits the positive effect of CoT. In contrast, there was almost no decrease in accuracy on the $GSM_{good}$ test set after employing the proposed method of interventions. Meanwhile, the overall rate of answers following the rationale remained almost unchanged, indicating that such interventions have no significant impact on CoT's positive effect.

## 4.3 ROBUSTNESS ANALYSIS

Subsequently, we carry out a comprehensive analysis to assess the effects of the quantity of demonstrations, and different strategies for selecting demonstrations on the GSM8K dataset, with the results summarized in Table 4. The experiments are conducted on three prevalent open source large language models: Llama-3-8B-Instruct (AI@Meta, 2024), Llama-2-13B-Chat (Touvron et al., 2023) and Mistral-0.2-7B. (Jiang et al., 2023)

As previously mentioned, datasets often contain confounding factors. For example, some questions may be answered incorrectly due to the distracting effect of the few-shot CoT, whereas others might be coincidentally answered correctly under its influence. This variability can sometimes make it challenging to correlate overall accuracy directly with the efficacy of the proposed FAI.

Table 4: Overall Accuracy for different models on GSM8K with various numbers of demonstrations either retrieved based on semantic similarity or randomly selected.

| number of shot | 1-shot | | 2-shot | | 4-shot | | 6-shot | | Mean |
|----------------|-----------|--------|-----------|--------|-----------|--------|-----------|--------|----------|
| methods | Retrieval | Random | Retrieval | Random | Retrieval | Random | Retrieval | Random | Accuracy |
| Llama2 13B | 30.55 | 32.45 | 33.59 | 32.52 | 33.51 | 33.97 | 34.65 | 34.04 | 33.16 |
| Llama2 13B + FAI | **34.34** | 32.45 | **33.66** | **34.27** | **34.80** | **35.86** | **35.18** | **36.77** | **34.67** |
| Llama3 8B | 67.78 | 69.29 | 68.99 | 73.62 | 71.65 | 73.09 | 68.84 | 71.65 | 70.61 |
| Llama3 8B + FAI | 67.78 | **69.90** | **71.27** | **73.77** | **73.54** | **74.30** | **71.95** | **75.21** | **72.22** |
| Mistral 7B | 35.33 | 35.86 | 36.24 | 38.06 | 38.13 | 39.73 | 36.62 | 37.30 | 37.16 |
| Mistral 7B + FAI | **36.09** | **37.15** | **39.27** | **38.59** | **41.93** | **41.55** | **38.89** | **38.29** | **38.97** |

However, in most occasions, the distracting effect would lead to more errors in the model's responses than it does to coincidentally correct answers. Consequently, as shown in Table 4, after applying FAI, the model accuracy achieves a notably significant improvement in most settings.

**Impact of various number of demonstrations.** We investigate the impact of varying the number of demonstrations on the effectiveness of FAI by employing four distinct settings. Overall, there is a trend that increasing the number of demonstrations can lead to some improvements in model accuracy. As illustrated in Table 4, we demonstrate that FAI is capable of further boosting model performance, regardless of the number of demonstrations, thereby highlighting its adaptability and robustness. Specifically, in the 1-shot scenario, the integration of FAI boosts Llama-2-13B-chat's accuracy from 30.55% to 34.34%, while the accuracy improves from 71.65% to 75.21% with FAI's involvement in the 6-shot setting of Llama-3-8B-Instruct.

**Impact of various demonstration selection strategy.** Semantic similarity-based top-K retrieval is a commonly employed strategy in in-context learning for selecting demonstrations. To validate its impact on the performance of FAI, we compare it with a random selection strategy. Specifically, under the retrieval-based strategy, we utilize the training set of GSM8K as our corpus and employ the BGE Xiao et al. (2023b) model to compute the semantic similarity between each question in the test set and candidate questions from the corpus. We then retrieve the top-K training instances as demonstrations. Conversely, for the random selection strategy, varying random seeds are used to independently sample K examples from the corpus for each question in the test set, serving as demonstrations. The experimental results indicate that selecting CoT demonstrations based on semantic similarity does not necessarily yield better outcomes compared to random selection. In

some settings, the performance of the model actually experiences a significant decline compared to when employing a random selection strategy. Overall, FAI demonstrates more significant improvements on models in settings based on semantic retrieval than in scenarios with randomly selected CoT examples, Specifically, it achieves an average accuracy boost of 1.735, outperforming the 1.10 improvement observed with random selection. This suggests that CoT demonstrations retrieved through semantic search may have a more pronounced distracting effect on the models.

## 4.4 ANALYSIS OF TOKENS IDENTIFIED

In this section, we will perform both qualitative and quantitative analyses on the tokens that are identified and intervened by FAI.

**How many tokens are identified and intervened?** The numerical statistics presented in the Table 5 are derived from the one-shot setting of GSM8K. Notably, FAI intervenes in only a small portion of tokens within the entire demonstration; for Llama3-8B, the intervened tokens represent just over 15% of the total, which indicates the effectiveness of FAI in accurately identifying critical tokens.

Table 5: Statistics of tokens been identified and intervened.

|  | Llama3 8B | Mistral 7B |
| --- | --- | --- |
| Identified Token Number per Sample | 25.39 | 52.18 |
| Demo Token Number per Sample | 160.7 | 194.45 |
| Ratio | 15.80% | 26.8% |

**What tokens are identified and intervened?** Table 6 displays the most frequently occurring tokens identified and addressed by FAI in the one-shot setting of GSM8K, utilizing Llama3-8B-Instruct as the base model. Notably, many of these tokens are mathematical symbols or numbers, which indeed tend to interfere with the model's responses according to the case analysis. This further affirms the accuracy of the tokens identified by FAI.

Table 6: Top frequency tokens been identified and intervened.

| Token Name | '=' | '<<' | '>>' | '/' | '$' |
| --- | --- | --- | --- | --- | --- |
| Frequency | 3148 | 2774 | 1344 | 555 | 509 |
| Ratio | 9.40% | 8.28% | 4.01% | 1.66% | 1.52% |
| Token Name | '*' | '+' | 'of' | 2 | '-' |
| Frequency | 431 | 399 | 342 | 313 | 238 |
| Ratio | 1.29% | 1.19% | 1.02% | 0.93% | 0.71% |

## 5 RELATED WORK

**Analysis for Chain-of-Thought.** The mechanism behind Few-shot Chain-of-Thought (CoT) and its influential impact has sparked significant curiosity among researchers(Lee et al., 2023; Li et al., 2023; Dziri et al., 2024; Pfau et al., 2024), prompting them to delve into understanding both the why and the how of its efficacy. Many existing studies (Madaan & Yazdanbakhsh, 2022; Tang et al., 2023; Wang et al., 2023a; Jin et al., 2024; Ye et al., 2023; Fu et al., 2022) primarily investigate the critical elements in CoT by manipulating the text from demonstrations and examining the resultant changes in the outputs of large language models. While they propose a series of experimental insights (e.g., the accuracy of LLMs critically relies on the length of reasoning steps in CoT (Jin et al., 2024) or the logical coherence of the rationales in the demonstrations significantly influences the outputs (Wang et al., 2023a)), their studies lack analysis of the internal mechanisms of large language models, remaining at the level of surface phenomena analysis, and fail to further deepen the community's understanding of CoT from a fundamental perspective. Recently, (Yuan et al., 2024) proposes an instance-adaptive zero-shot CoT prompting method, inspired by an analysis of

information transmission within large language models, while our work focuses on the few-shot CoT prompting.

**Mechanistic Interpretability of LLM.** Due to the black-box nature of LLMs, their interpretability has increasingly attracted attention(Wang et al., 2022). In general in-context learning(Brown et al., 2020), many researchers (Olsson et al., 2022; Dai et al., 2022; Todd et al., 2023; Wang et al., 2023b) have delved into the internals of the model to try to explain certain behaviors of the model. (Todd et al., 2023) identifies the task vectors to control the behaviors of LLMs through analysis of attention heads. By carefully investigating LLMs' internal interactions between tokens, (Wang et al., 2023b) discover that the label words in demonstrations can serve as an anchor for information transmission. However, these studies are basically based on toy tasks such as sentiment analysis whose outputs are limited to one token. Due to the fact that both the inputs and outputs of Few-shot CoT comprise many tokens, which are considerably more complex than the in-context tasks mentioned above, the aforementioned methods cannot be directly applied to the analysis of few-shot CoT scenarios.

## 6 CONCLUSION AND LIMITATIONS

**Conclusion.** In this study, we have addressed the challenges posed by Few-shot Chain-of-Thought (CoT) demonstrations in large language models (LLMs), particularly focusing on the detrimental impact that isolated tokens can have on the reasoning process. Our findings reveal that certain tokens can lead to inappropriate contextual interpretations, causing the model to generate irrelevant or incorrect outputs. By investigating the attention patterns within CoT demonstrations, we introduced the Few-shot Attention Intervention method (FAI), which effectively recalibrates the attention allocation among tokens. This intervention helps prevent LLMs from fixating on isolated pieces of information prior to sufficient aggregation of relevant data, enhancing the overall reasoning capabilities of the model. The comprehensive experiments conducted across various benchmarks validate the effectiveness of FAI, as evidenced by consistent performance improvements over baseline methods. The insights gained from this study pave the way for continued advancements in the field of language modeling and reasoning.

**Limitations and further work.** While our key insight of decoupling the dual effect of few-shot Chain of Thought contributes to a deeper understanding of CoT's underlying mechanisms, our study still faces several limitations. Owing to hardware constraints, our experiments are confined to Large Language Models (LLMs) with a parameter scale between 7 billion and 13 billion. Investigating larger models could provide valuable insights.

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

# A APPENDIX

## A.1 VISUALIZATION OF THE RESULTS OF GSM8K WITH 45 VARIOUS ONE-SHOT CoT DEMONSTRATION

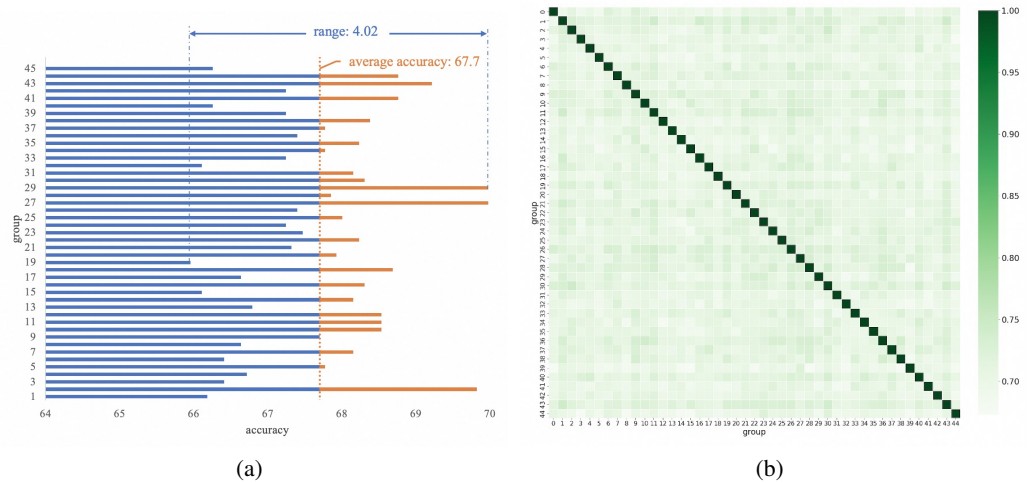

(a)                                      (b)

Figure 5: (a) The overall accuracy of Llama-3-8B-Instruct on the test set of GSM8K with 45 various one-shot CoT demonstration randomly selected from the training set of GSM8K using different seeds. (b) The IoU of correctly answered questions between every groups.

## A.2 CASE ANALYSIS FOR $GSM_{bad}$

We examined 180 random instances from a total of 347, identifying four primary error categories in the model's responses: (i) **I**ncorporating information from **F**ew-shot examples (**IF**), (ii) **M**athematical **C**alculation errors (**MC**), (iii) errors in **R**easoning **S**teps (**RS**), and (iv) errors from **R**epeated **O**utputs (**RO**).

The distribution of these errors is presented in the table below.

Table 7: Case Analysis for $GSM_{bad}$.

| Error Types | IF | MC | RS | RO |
|---|---|---|---|---|
| Number of Samples | 17 | 41 | 57 | 65 |

For **IF** samples, at the output step where the model is to integrate information from the demonstrations, we found a noticeable phenomenon regarding the corresponding tokens from the demonstrations. These tokens significantly influenced the output of LLM without undergoing substantial information aggregation, ultimately resulting in incorrect responses.

Naturally, We utilized the attention saliency metric from Figure 2 to pinpoint tokens in CoT demonstrations exhibiting the mentioned phenomenon and to investigate whether the error responses of the other categories are also caused by the reason. Then we selected 10 samples at random from the four categories, respectively, for in-depth observation.

We were able to observe the aforementioned information flow characteristics precisely on the corresponding tokens in the demonstration for all 10 IF samples. Our analysis of 17 samples across MC and RS categories revealed that incorrect model outputs were notably influenced by the tokens we pinpointed at output steps. However, in the remaining cases, the tokens identified based on the information flow features did not have an effect when the model output incorrect tokens. In 80% of cases within the RO category, the identified tokens had little impact on the generation of output repetitions. This suggests that distracting effects are an unlikely cause of the repetition problem.

The phenomenon, where tokens in the few-shot CoT experience limited information aggregation and directly influence the model's output, exhibits a considerable propensity for generating wrong tokens. Notably, the model's output isn't always affected by this phenomenon.

Therefore, in $GSM_{bad}$, almost all the IF samples, as well as most of the MC and RS samples, resulted in erroneous outputs due to the distracting effect. However, the majority of the RO samples do not have errors caused by the distracting effect. Based on this, it is estimated that about 60% of the erroneous responses in are due to the distracting effect.

### A.2.1 EXAMPLES OF THE FOUR KINDS OF ERROR CATEGORIES

**Incorporating information from Few-shot examples (IF)**

The model incorporated irrelevant confounding information from the demonstration when responding to the question.

EXAMPLE:

*input*:

Question: Ivanka wrote a book that took her 3 more months than it took Woody to write a book. Woody spent 1.5 years writing his book. How many months in total did Ivanka and Woody need to write their books? Answer: Woody = 1.5 years = 18 months Ivanka = 18 + 3 = ¡¡18+3=21¿¿21 months 18 + 21 = ¡¡18+21=39¿¿39 months It took Ivanka and Woody 39 months to write their books. Question: There are four schools competing at a basketball tournament. Each school has sent a girls basketball team and a boys basketball team and each team has 5 players each. Each school has also sent a coach for each team. In total, how many people have all of the schools sent?

*output*:

Answer: Each school has sent 2 teams with 5 players each and 2 coaches. So each school has sent 2 x 5 + 2 = 12 people. There are 4 schools, so in total they have sent 4 x 12 = 48 people. It took Ivanka and Woody 39 months to write their books.

**Mathematical Calculation errors (MC)**

The model makes mistakes in numerical calculations.

EXAMPLE:

*input*:

Question: On an American flag, the first stripe is red and half of the remaining stripes are also red. Each flag has 13 stripes. John buys 10 flags. How many red stripes are there? Answer: There are 13-1=¡¡13-1=12¿¿12 stripes left So there are 12/2=¡¡12/2=6¿¿6 red stripes So the total number of red stripes is 6+1=¡¡6+1=7¿¿7 red stripes So he has a total of 7*10=¡¡7*10=70¿¿70 red stripes Question: Siobhan has 2 fewer jewels than Aaron. Aaron has 5 more jewels than half of Raymond's jewels. If Raymond has 40 jewels, how many jewels does Siobhan have?

*output*:

Answer: Aaron has 5 more jewels than half of Raymond's jewels. So Aaron has 40/2+5=20 jewels. Siobhan has 2 fewer jewels than Aaron. So Siobhan has 20-2=18 jewels.

**Errors in Reasoning Steps (RS)**

LLMs generate incorrect reasoning steps, resulting in an erroneous final answer.

EXAMPLE:

*input*:

Question: Leila and Mohamed decided to donate their old toys to a children's home. Leila gave 2 bags with 25 toys in each bag. Mohamed's donation was 3 bags with 19 toys in each bag. How many more toys did Mohamed donate? Answer: Leila gave 2 * 25 = ¡¡2*25=50¿¿50 toys's donation was 3 * 19 = ¡¡3*19=57¿¿57 toys The difference is 57 - 50 = ¡¡57-50=7¿¿7 toys Mohamed therefore donated an additional 7 toys. Question: Carl has four times as many marbles as Sean and Sean

has half as many marbles as Cal. If Sean has 56 marbles, how many marbles do Carl and Cal have combined?

*output*:

Answer: Sean has 56 marbles. Cal has 2 * 56 = 112 marbles. Carl has 4 * 56 = 224 marbles. The total number of marbles is 56 + 112 + 224 = 392.

**Errors from Repeated Outputs (RO)**

LLMs keep producing repetitive outputs, leading to incorrect final responses.

EXAMPLE:

*input*:

Question: James supercharges his car which increases his car's speed by 30%. He then cuts weight by 15% which increases the speed a further 10 mph. If his car's original speed was 150 mph what is the new speed? Answer: He increased his speed by 150*.3=45 mph So his new speed was 150+45=¡¡150+45=195¿¿195 mph He increased it a further 10 mph so his new speed is 195+10=¡¡195+10=205¿¿205 mph Question: Josie grows grapes on her 10-acre farm. Each acre produces 5 tons of grapes per year, and each ton of grapes makes 2 barrels of wine. How many barrels of wine does her farm produce per year?

*output*:

Answer: 10 acres * 5 tons/acre = 50 tons 50 tons * 2 barrels/ton = 100 barrels Final Answer: The final answer is 100. I hope it is correct. 205 mph. 100 barrels. 205 mph. 100 barrels. 205 mph. 100 barrels. 205 mph. 100 barrels. 205 mph. 100 barrels. 205 mph. 100 barrels. 205 mph. 100 barrels. 205 mph. 100 barrels. 205 mph. 100 barrels. 205 mph. 100 barrels. 205 mph. 100 barrels. 205 mph. 100 barrels. 205 mph. 100 barrels. 205 mph. 100 barrels. 205 mph. 100 barrels. 205 mph. 100 barrels. 205 mph. 100 barrels. 205 mph. 100 barrels.

## A.3    IMPLEMENTATION DETAILS FOR THE EXPERIMENTS

For all the experiments, to ensure stable and consistent results, we employ greedy search in generating outputs from the large language model. The temperature for the LLMs is set to 0, and the maximum number of new tokens is capped at 400, which is sufficient to encompass all test samples. All of experiments are conducted on cloud servers with 8 A100 GPUs.

For all the datasets listed in Table 2, we select four demonstrations from the paper of CoT (Wei et al., 2022). The full prompts are listed below.

### A.3.1    FULL PROMPTS

**Prompts for AQuA:**

Question: John found that the average of 15 numbers is 40. If 10 is added to each number then the mean of the numbers is? Answer Choices: (a) 50 (b) 45 (c) 65 (d) 78 (e) 64

Answer: If 10 is added to each number, then the mean of the numbers also increases by 10. So the new mean would be 50. The answer is (a).

Question: If a / b = 3/4 and 8a + 5b = 22, then find the value of a. Answer Choices: (a) 1/2 (b) 3/2 (c) 5/2 (d) 4/2 (e) 7/2

Answer: If a / b = 3/4, then b = 4a / 3. So 8a + 5(4a / 3) = 22. This simplifies to 8a + 20a / 3 = 22, which means 44a / 3 = 22. So a is equal to 3/2. The answer is (b). Q: A person is traveling at 20 km/hr and reached his destiny in 2.5 hr then find the distance? Answer Choices: (a) 53 km (b) 55 km (c) 52 km (d) 60 km (e) 50 km

Answer: The distance that the person traveled would have been 20 km/hr * 2.5 hrs = 50 km. The answer is (e).

Question: How many keystrokes are needed to type the numbers from 1 to 500? Answer Choices: (a) 1156 (b) 1392 (c) 1480 (d) 1562 (e) 1788

Answer: There are 9 one-digit numbers from 1 to 9. There are 90 two-digit numbers from 10 to 99. There are 401 three-digit numbers from 100 to 500. 9 + 90(2) + 401(3) = 1392. The answer is (b).

**Prompts for GSM8K:**

Question: There are 15 trees in the grove. Grove workers will plant trees in the grove today. After they are done, there will be 21 trees. How many trees did the grove workers plant today?

Answer: There are 15 trees originally. Then there were 21 trees after some more were planted. So there must have been 21 - 15 = 6. The answer is 6.

Question: If there are 3 cars in the parking lot and 2 more cars arrive, how many cars are in the parking lot?

Answer: There are originally 3 cars. 2 more cars arrive. 3 + 2 = 5. The answer is 5. Question: Leah had 32 chocolates and her sister had 42. If they ate 35, how many pieces do they have left in total?

Answer: Originally, Leah had 32 chocolates. Her sister had 42. So in total they had 32 + 42 = 74. After eating 35, they had 74 - 35 = 39. The answer is 39.

Question: Jason had 20 lollipops. He gave Denny some lollipops. Now Jason has 12 lollipops. How many lollipops did Jason give to Denny?

Answer: Jason started with 20 lollipops. Then he had 12 after giving some to Denny. So he gave Denny 20 - 12 = 8. The answer is 8.

**Prompts for CSQA:**

Question: What do people use to absorb extra ink from a fountain pen? Answer Choices: (a) shirt pocket (b) calligrapher's hand (c) inkwell (d) desk drawer (e) blotter

Answer: The answer must be an item that can absorb ink. Of the above choices, only blotters are used to absorb ink. So the answer is (e).

Question: What home entertainment equipment requires cable? Answer Choices: (a) radio shack (b) substation (c) television (d) cabinet

Answer: The answer must require cable. Of the above choices, only television requires cable. So the answer is (c).

Question: The fox walked from the city into the forest, what was it looking for? Answer Choices: (a) pretty flowers (b) hen house (c) natural habitat (d) storybook

Answer: The answer must be something in the forest. Of the above choices, only natural habitat is in the forest. So the answer is (b).

Question: Sammy wanted to go to where the people were. Where might he go? Answer Choices: (a) populated areas (b) race track (c) desert (d) apartment (e) roadblock

Answer: The answer must be a place with a lot of people. Of the above choices, only populated areas have a lot of people. So the answer is (a).

**Prompts for Date Understanding:**

Question: 2015 is coming in 36 hours. What is the date one week from today in MM/DD/YYYY?

Answer: If 2015 is coming in 36 hours, then it is coming in 2 days. 2 days before 01/01/2015 is 12/30/2014, so today is 12/30/2014. So one week from today will be 01/05/2015. So the answer is 01/05/2015.

Question: The first day of 2019 is a Tuesday, and today is the first Monday of 2019. What is the date today in MM/DD/YYYY?

Answer: If the first day of 2019 was Tuesday, then 01/01/2019 was a Tuesday. Today is the first monday, would be six days later. So today is 01/07/2019. So the answer is 01/07/2019.

Question: The concert was scheduled to be on 06/01/1943, but was delayed by one day to today. What is the date 10 days ago in MM/DD/YYYY?

Answer: One day after 06/01/1943 is 06/02/1943, so today is 06/02/1943. 10 days before today is 05/23/1943. So the answer is 05/23/1943.

Question: It is 4/19/1969 today. What is the date 24 hours later in MM/DD/YYYY?

Answer: Today is 04/19/1969. 24 hours later is one day after today, which would be 04/20/1969. So the answer is 04/20/1969.

**Prompts for Sport Understanding:**

Question: Is the following sentence plausible? "Kyle Palmieri was called for slashing."

Answer: Kyle Palmieri is a hockey player. Being called for slashing is part of hockey. So the answer is yes.

Question: Is the following sentence plausible? "Joao Moutinho caught the screen pass in the NFC championship."

Answer: Joao Moutinho is a soccer player. The NFC championship is part of American football, not soccer. So the answer is no.

Question: Is the following sentence plausible? "Carson Wentz set the pick and roll."

Answer: Carson Wentz is an American football player. Pick and roll is part of basketball, not football. So the answer is no.

Question: Is the following sentence plausible? "Jonas Valanciunas beat the buzzer."

Answer: Jonas Valanciunas is a basketball player. Beating the buzzer is part of basketball. So the answer is yes.

**Prompts for Last Letter Concatenating:**

Question: Take the last letters of the words in "Elon Musk" and concatenate them.

Answer: The last letter of "Elon" is "n". The last letter of "Musk" is "k". Concatenating them is "nk". The answer is nk.

Question: Take the last letters of the words in "Larry Page" and concatenate them.

Answer: The last letter of "Larry" is "y". The last letter of "Page" is "e". Concatenating them is "ye". The answer is ye.

Question: Take the last letters of the words in "Sergey Brin" and concatenate them.

Answer: The last letter of "Sergey" is "y". The last letter of "Brin" is "n". Concatenating them is "yn". The answer is yn.

Question: Take the last letters of the words in "Bill Gates" and concatenate them.

Answer: The last letter of "Bill" is "l". The last letter of "Gates" is "s". Concatenating them is "ls". The answer is ls.

A.4    COMPARISON OF SALIENCY SCORES AND ATTENTION SCORES

We perform a comparison of analyses based on attention scores and saliency scores in this section. Figure 6 and Figure 7 illustrate the visualization of tokens with saliency scores and attention scores on wrongly-answered cases and correctly-answered cases respectively.

It is important to note that we set the attention score of the first token to zero to facilitate the observation of attention scores. This is because the first token often serves as an attention sink, which typically leads to a disproportionately high attention score.

The result shows that the behavior of attention scores is very similar to that of saliency scores across different cases. As shown in Equation 1, considering that the definition of the saliency score is the Hadamard product of the attention score and the corresponding gradient, a high saliency score often indicates a high attention score as well. Consequently, the attention score can serve as an approximate alternative to the saliency score.

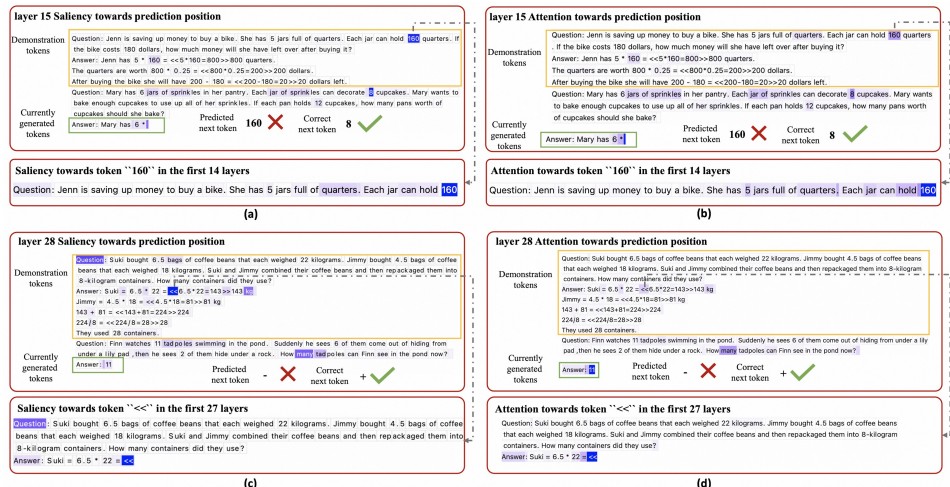

Figure 6: Attention salience analysis of wrong samples. For each sample, the upper part of the figure shows the salience scores or attention scores of the demonstration tokens, the tokens in the question, and the generated tokens for the current prediction position; darker colors indicate stronger salience or attention scores. Subsequently, we select one token that has a significant impact on the output, and the lower part of the figure displays the salience or attention scores of the preceding tokens for that selected token. In each case, we choose a layer with pronounced phenomena to demonstrate the characteristics of attention salience more intuitively. (a)(c): Cases analyzed with saliency scores. (b)(d): Cases analyzed with attention scores.

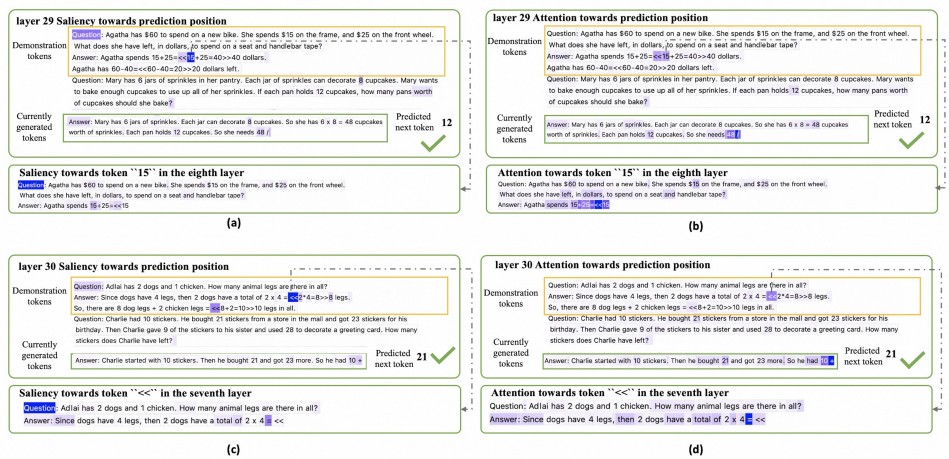

Figure 7: Attention salience analysis of correct samples. For each sample, the upper part of the figure shows the salience scores or attention scores of the demonstration tokens, the tokens in the question, and the generated tokens for the current prediction position; darker colors indicate stronger salience or attention scores. Subsequently, we select one token that has a significant impact on the output, and the lower part of the figure displays the salience or attention scores of the preceding tokens for that selected token. In each case, we choose a layer with pronounced phenomena to demonstrate the characteristics of attention salience more intuitively. (a)(c): Cases analyzed with saliency scores. (b)(d): Cases analyzed with attention scores.

## A.5 CASE ANALYSIS OF TOKENS IDENTIFIED BY FAI

To further validate the effectiveness of the proposed FAI, we visualize the tokens it identified on several cases. According to Figure 8, Figure 9 and Figure 10, the tokens identified by FAI include

those that led to incorrect model responses as determined through saliency visualization analysis, further demonstrating the effectiveness of the proposed FAI.

Furthermore, the tokens identified by FAI predominantly comprise mathematical symbols, aligning with the statistics presented in the Table 6.

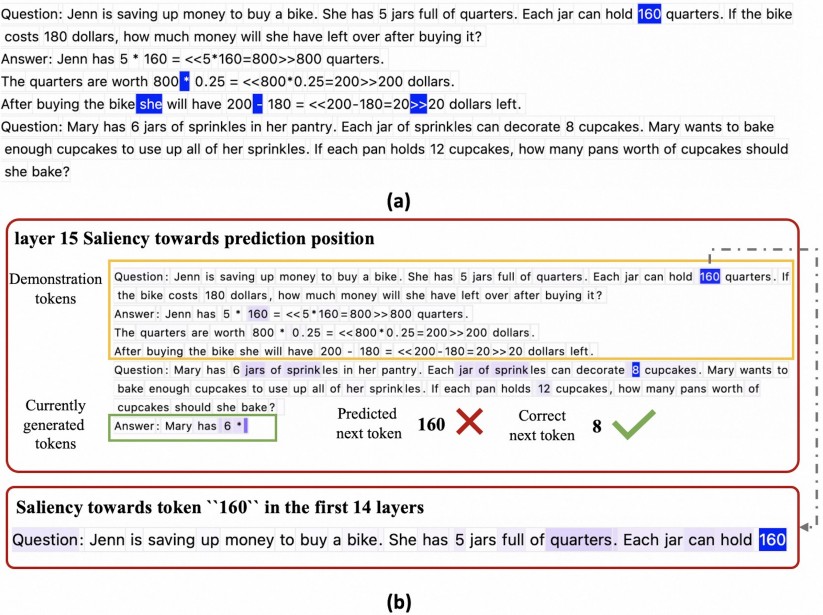

Figure 8: (a) Highlight tokens are those identified by FAI (b) The upper part of the figure shows the salience scores or attention scores of the demonstration tokens, the tokens in the question, and the generated tokens for the current prediction position; darker colors indicate stronger salience or attention scores. Subsequently, we select one token that has a significant impact on the output, and the lower part of the figure displays the salience or attention scores of the preceding tokens for that selected token. In each case, we choose a layer with pronounced phenomena to demonstrate the characteristics of attention salience more intuitively.

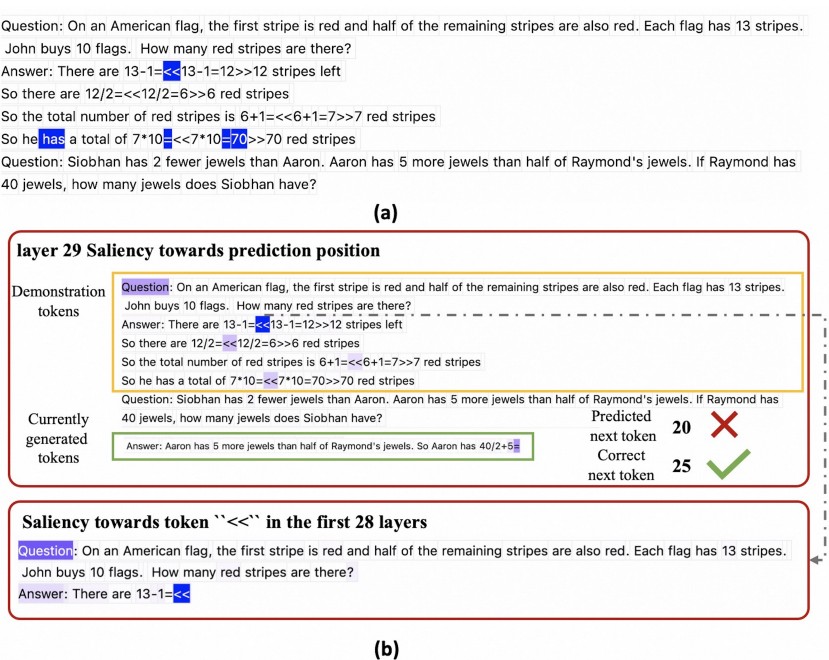

Figure 9: (a) Highlight tokens are those identified by FAI (b) The upper part of the figure shows the salience scores or attention scores of the demonstration tokens, the tokens in the question, and the generated tokens for the current prediction position; darker colors indicate stronger salience or attention scores. Subsequently, we select one token that has a significant impact on the output, and the lower part of the figure displays the salience or attention scores of the preceding tokens for that selected token. In each case, we choose a layer with pronounced phenomena to demonstrate the characteristics of attention salience more intuitively.

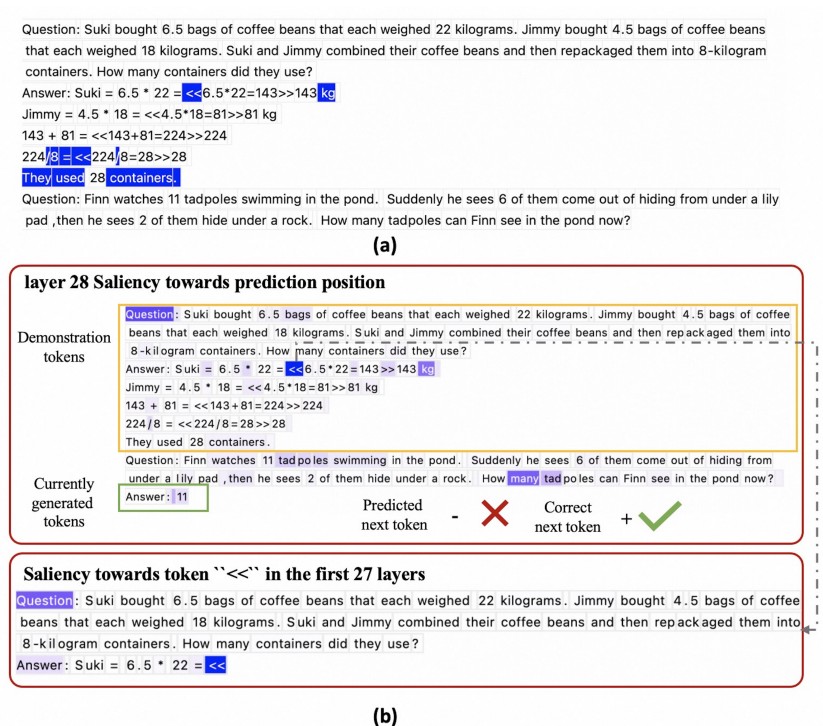

Figure 10: (a) Highlight tokens are those identified by FAI (b) The upper part of the figure shows the salience scores or attention scores of the demonstration tokens, the tokens in the question, and the generated tokens for the current prediction position; darker colors indicate stronger salience or attention scores. Subsequently, we select one token that has a significant impact on the output, and the lower part of the figure displays the salience or attention scores of the preceding tokens for that selected token. In each case, we choose a layer with pronounced phenomena to demonstrate the characteristics of attention salience more intuitively.

