# OpenReview forum: "Don't Take Things Out of Context: Attention Intervention for Enhancing Chain-of-Thought Reasoning in Large Language Models"
_ICLR.cc/2025/Conference — ICLR 2025 Poster_

### Official Review · Reviewer_aVVN · 2024-10-27

**Soundness:** 3
**Presentation:** 2
**Contribution:** 3
**Rating:** 8
**Confidence:** 4

**Summary:**

The paper explores the issue of localized distractions within CoT in LLMs. While CoT enhances LLM reasoning by guiding models through structured reasoning steps, isolated tokens or phrases in the demonstrations can mislead the model, causing it to focus excessively on irrelevant details, which disrupts its reasoning process. To mitigate this, the authors propose a Few-shot Attention Intervention (FAI) method. Through experiments on multiple reasoning benchmarks, FAI demonstrates consistent improvements,  indicating its effectiveness in enhancing CoT robustness without extensive computational overhead​.

**Strengths:**

1. The idea is novel, and the research question is interesting and valuable.
2. The paper is clearly structured, with fairly extensive experiments and the results show good potential.

**Weaknesses:**

1. The paper citation format is improper, and \citet{} and \citep{} are not used appropriately.
2. The experimental settings are not clear (especially the hyperparameter settings of the model, like decode strategy, max_new_token), which is not conducive to reproduction.

**Questions:**

I noticed that all your experiments were conducted on cloud servers with 8 A100 GPUs, so why did you only evaluate 7B~13B models?

---

> ### Author Response · Authors · 2024-11-22
> **Response to Reviewer aVVN**
>
> Thanks a lot for your valuable reviews, and we appreciate the time and effort you have taken. We will address your concerns as outlined below.
>
> **Weakness 1:**
>
> The paper citation format is improper, and \citet{} and \citep{} are not used appropriately.
>
> **Response to Weakness 1:**
>
> We apologize for the errors we made and have thoroughly reviewed the entire paper, correcting all citations.
>
> **Weakness 2:**
>
> The experimental settings are not clear (especially the hyperparameter settings of the model, like decode strategy, max_new_token), which is not conducive to reproduction.
>
> **Response to Weakness2:**
>
> We apologize for any confusion. All our experiments utilize a greedy search method to generate outputs from the large language model, as outlined in Appendix A.3. The temperature for the LLM is set to 0, and the maximum number of new tokens is capped at 400, which is sufficient to encompass all test samples.
>
> We have added more details about the experimental settings in the paper and we are planning to release our code to enhance reproducibility.
>
>
> **Question 1:**
>
> I noticed that all your experiments were conducted on cloud servers with 8 A100 GPUs, so why did you only evaluate 7B~13B models?
>
> **Response to Question 1:**
>
> We apologize for any confusion. Indeed, our cloud server operates on a shared task queuing basis among multiple users, which limits our access to sufficient computing power. Recently, we have conducted additional experiments using the Llama-3-70B-Instruct model, and the results are presented in the table below:
>
> |  method  | AQuA  | GSM8K | CSQA | Date | Sport | Last Letter
> |---|---|---|---|---|---|---|
> | Llama3-8B | 40.94 | 70.32 | 71.17 | 64.00 | 95.60 | 58.67 |
> | Llama3-8B + FAI | 46.85 | 71.24 | 74.28 |  65.60 | 96.00 | 62.00 |
> | Llama3-70B | 66.14 | 91.28 | 77.31 | 87.60 | 97.2 | 84.00 |
> | Llama3-70B +FAI | 66.53 | 91.28 | 78.62 | 88.0 | 98.0 | 85.33 |
>
> Llama-3-70B-Instruct demonstrates strong performance across various benchmarks. However, applying FAI to this already powerful baseline consistently enhances its results across almost all benchmarks, further highlighting the effectiveness of FAI.

---

> > ### Comment · Reviewer_aVVN · 2024-11-24
> > **Official Comment**
> >
> > Thanks for the clarification. I've increased my score.

---

> ### Author Response · Authors · 2024-11-24
> **Inquiry Regarding Rebuttal Feedback**
>
> Dear Reviewer aVVN,
>
> We sincerely appreciate your time and effort in reviewing our manuscript and offering valuable suggestions. We provided detailed clarifications in response to your questions a few days ago. If you have any additional feedback, concerns, or questions regarding our response, we would greatly appreciate hearing from you and welcome further discussion.

---

### Official Review · Reviewer_k9ai · 2024-11-01

**Soundness:** 2
**Presentation:** 2
**Contribution:** 2
**Rating:** 6
**Confidence:** 3

**Summary:**

This paper proposes FAI that dynamically analyzes the attention patterns of demonstrations to accurately identify these tokens, followed by targeted adjustments to the attention weights to effectively suppress their distracting impact on LLMs. Comprehensive experiments across various benchmarks validate the effectiveness of FAI.

**Strengths:**

1.	The proposed FAI offers a new perspective for analyzing the impact of few-shot CoT on responses generated by LLMs. It enables LLMs to focus more on global information rather than individual tokens.

2.	The motivation is illustrated with examples to aid reader understanding. The method is clearly described, and the experiments are extensive in scope.

**Weaknesses:**

1. FAI is kind of incremental. It modifies the attention weight matrix to intervene in information aggregation, thereby improving the effect of CoT. However, the method is largely built on the single-step saliency scoring approach like Wang et al. (2023b) and simply identifies a threshold to block the information flow between the layers in the LLMs by attention, this is also done in the traditional pre-trained language models. However attention-based saliency scoring is also widely discussed by existing studies [1]. In this way, the paper offers limited innovation.

2. The experimental analysis is insufficient and should be validated on more models. Please see the fourth question below.

3. The description of details needs further refinement which can be seen in the questions below.

[1] From Understanding to Utilization: A Survey on Explainability for Large Language Models.

**Questions:**

Major Questions
1. Compared with the method proposed by Wang et al. (2023b) that introduces a learnable parameter to dynamically modify the attention weight matrix, what are the advantages and innovations of this method? The authors claim that “while the saliency score emerges … alternatives.” from line 246 to 248. Can you explain more about this or conduct an analysis of the complexity comparison between these two? BTW, how to identify the hyper-parameter $\lambda$ in the method?

2. At line 134, the author mentions that existing single-step methods may overlook crucial information, and the proposed method has more advantages due to its ability to dynamically adjust the attention weight matrix at each step. However, these baselines were not compared in the experimental part. Experimental analyses need to be given that the proposed method is better than single-step ones. In addition, since this method modifies the attention weight matrix at each step, it seems to be more time-consuming than single-step methods, so relevant experimental analysis also needs to be provided.

3. The figures are blurry. For example, in Figure 2, why do the positive examples (a) and (c) use the previous layer to indicate significance, while the negative examples (b) and (d) include all previous layers? This seems unfair. Would the first 30 layers in (a) achieve the same effect as in (b) and (d)? And would the 14th layer in (b) produce a similar effect to (a) and (c)? A more thorough explanation is needed here.

4. How about we apply FAI to larger models such as Llama-3-70B? Or LLMs in other series such as GPT2-XL (1.5 Billion parameter) and GPT-NEO-2.7B? The experiments are not sufficient and should be verified on more types of models with different parameters.

5. The author selects samples with an accuracy rate greater than 90% as GSM_bad, but lacks reasons. Is there some experiment or analysis that can support the idea that the errors in them are more likely to be caused by the distracting effect? I think visualization towards the FAI attention is helpful.

6. What are the limitations of FAI? Even though the authors showcase some examples of the four error categories in A.2.1, I am interested in what examples it tends to answer incorrectly that could otherwise be answered correctly without FAI? In addition, what examples does it tend to answer incorrectly even with FAI?

Minor Questions
1.	Why does the attention weight matrix in Figure 3 exclude the query question in Layer n but include it in Layer n+1?

2.	The description on lines 273 to 274 contradicts the description on lines 260 to 263.

3.	There is no explanation of Table 3.

4.	How is RAFR calculated?

5.	What does “contrastive” in Figure 4 stand for?

---

> ### Author Response · Authors · 2024-11-22
> **Response to Reviewer k9ai (1/6)**
>
> We sincerely appreciate the time and effort you dedicated to reviewing our paper. Below are our responses.
>
> **Weakness 1:**
>
> The proposed method is largely built on the single-step saliency scoring approach like Wang et al. (2023b) and simply identifies a threshold to block the information flow between the layers in the LLMs by attention, this is also done in the traditional pre-trained language models.
>
> **Response to Weakness 1:**
>
> As you mentioned, attention saliency is a widely used analysis tool that has inspired the work of Wang et al. (2023b) and others. We would like to emphasize that, as far as we know, we are the first to apply attention saliency to the more complex scenario of few-shot CoT. Our in-depth analysis offers valuable insights that can significantly enhance the capabilities of few-shot CoT.
>
> Attention saliency is a well-established technique for interpreting and analyzing model behavior, with its origins traceable to [2], published in NeurIPS 2019. Notably, Wang et al. (2023b) represent the first study to apply the saliency technique outlined in [2] to investigate the patterns of in-context learning (ICL) in language models. Their key contribution is the discovery that label words function as anchors, aggregating and distributing information during ICL.
>
> In addition to Wang et al. (2023b), several other works have built upon saliency scores to analyze specific tasks or areas. For instance, [3] employs a similar information flow approach to interpret and address knowledge conflicts within the internal memory of large language models (LLMs) and the external context provided.
>
> As far as we know, we are the first to perform saliency analysis on the topic of few-shot CoT. The primary contribution of our paper lies in the exploration and analysis of the interference phenomena associated with few-shot CoT prompting on model responses—a topic that has received limited attention in prior research. As noted in lines 135 to 137, analyzing saliency scores in the context of few-shot CoT can be challenging. The CoT demonstrations do not directly impact the model's final output; rather, they influence the answer indirectly by shaping how the model generates its rationale. Consequently, relying solely on single-step approaches makes it difficult to identify failure cases in few-shot CoT.
>
> From our exploration of few-shot CoT, we find that a token in the demonstration that directly assigns a high saliency score to the model's predicted position tends to disrupt the model's output until this token receives significant information aggregation from other tokens at any given layer.
>
> In a word, saliency scores serve merely as a tool to analyze the behavior of few-shot CoT.  The innovations of our work compared to previous efforts are reflected in:
> 1. the complexity of the scenarios addressed: our work resides in applying this approach to the intricate challenge of few-shot CoT tasks,  while Wang et al. (2023b) focus on ICL single-token classification tasks. As mentioned before,  without a deep analysis of the dynamic rationale generation process of CoT, it is difficult to accurately identify its failure modes, which is far more complex than simply generating a single token in an ICL task.
>
> 2. a novel and different understanding of the underlying mechanisms generated by few-shot CoT analysis: Wang et al. (2023b) discover that label words function as anchors, while our work focus on the failure mode of few-shot CoT, analyzing the demonstration examples will affect CoT's reasoning dynamic by distracting models with specific tokens. As described by all reviewers, our findings offer novel insight into understanding the few-shot CoT dynamic.
>
> [2] Michel et al. Are sixteen heads really better than one?
>
> [3] Jin et al. Cutting Off the Head Ends the Conflict: A Mechanism for Interpreting and Mitigating Knowledge Conflicts in Language Models
>
> **Weakness 2 & Weakness 3:**
>
> The experimental analysis is insufficient and should be validated on more models. Please see the fourth question below.
>
> The description of details needs further refinement which can be seen in the questions below.
>
> **Response to Weakness 2 & Weakness 3:**
>
> Thanks for the valuable comments. We have conducted additional experiments and analyses as you suggested. We will provide detailed responses to the corresponding questions below.

---

> ### Author Response · Authors · 2024-11-22
> **Response to Reviewer k9ai (2/6)**
>
> **Major Question 1:**
>
> - Compared with the method proposed by Wang et al. (2023b)
> - analysis of the complexity comparison
> - how to identify the hyper-parameter in the method?
>
> **Response to Major Question 1 :**
>
> Thanks for the comment. We will respond to the three questions above in order.
>
> ***Comparison with Wang et al. (2023b)***
>
> We have carefully examined the method proposed by Wang et al. (2023b). There are significant differences between our approach and theirs in the following aspects:
>
> #### The two methods are applicable to different types of tasks.
>
> This method is specifically designed for *classification tasks that produce a single output token*, such as sentiment analysis, enabling the use of a classification loss to train this learnable parameter. For *open-ended real-world natural language tasks*, such as CoT reasoning, this method is not able to work. However, the FAI approach we propose is not limited to any specific type of task and does not require any task-specific training. It only requires a small cost during the inference phase, enabling further improvements over few-shot CoT prompting.
>
> #### The principles and mechanisms underlying the two methods differ significantly.
>
> This approach is based on their observation of the information flow patterns in in-context learning: label words serve as anchors that aggregate and distribute information in ICL. A crucial prerequisite for the effectiveness of this method is that information from the demonstrations converges onto the label words within the model's shallow layers. Therefore, by learning a parameter, they can reweight the attention from these label words toward the model's prediction positions, enabling a dynamic adjustment of the contributions from various demonstrations.
>
> Nevertheless, our topic focuses on the potential distracting effect of few-shot CoT. The proposed FAI method is built upon our observation that tokens which haven’t undergo significant information aggregation in the CoT demonstrations are more likely to disrupt the generation of rationales. Naturally, by blocking the direct information transfer from tokens with such pattern to the prediction position of models brings effective enhancement of performance.
>
> Consequently, our analysis is grounded in distinct scenarios and mechanisms when compared to Wang's study, offering valuable insights within their respective contexts of ICL with single token classification and few-shot CoT.
>
> ***complexity comparison between saliency score and attention weight***
>
> Compared to a normal generation, running FAI using saliency score requires the following additional steps:
> 1. Generate a single token and calculate its prediction loss.
> 2. Record the attention weight matrix of each head.
> 3. Loss back propagation.
> 4. Multiply the gradient of the attention weight matrix for each head by the corresponding attention weight to derive the saliency score.
> 5. Locate the tokens within the demonstration where the saliency scores suggest that substantial information aggregation did not occur.
> 6. Delete the generated tokens and regenerate them. While doing so, set the attention scores of the tokens in the specified position to zero in order to minimize their influence on the predicted token.
> 7. Repeat the above steps for each generated token
>
> The additional steps to run FAI using attention weights are as follows:
> 1. Input the CoT demonstrations into the model separately, run a single forward pass, and record the attention matrices for each head.
> 2. Based on the attention scores, identify the tokens in the demonstrations that did not exhibit significant aggregation, and record their corresponding layers and positions.
> 3. Input the CoT demonstrations and the question into the model. During the forward pass, set the attention scores to zero for the corresponding positions recorded in step 2 when generating each token.
>
> The operations in step 1 and step 2 need to be performed only once for a given problem. The additional computational cost introduced by step 3 on top of the existing generation is negligible. As a result, the computational cost of running FAI with the attention weights remains very low. *As highlighted by reviewer wLEA, the method does not bring extra computation gains (only a constant computational overhead)*.
>
> ***How to identify the hyper-parameter in the method***
>
> The hyper-parameter λ is used to define a threshold τ to determine whether the information aggregation of a token is significant.
>
> Therefore, a larger λ will involve less tokens to be intervened while a smaller λ identifies more tokens.
> In the experiments outlined in the paper, we did not intentionally adjust the value of λ.
>
> Given that the aggregation coefficient α ranges from 0 to 1, we set λ to 1. This choice ensures that the threshold τ for different tokens remains within the range of 0 to 1. Our experimental results indicate that this hyper-parameter value exhibits good generalization performance.

---

> ### Author Response · Authors · 2024-11-22
> **Response to Reviewer k9ai (3/6)**
>
> **Major Question 2:**
>
> - Comparison with single-step methods
> - In addition, since this method modifies the attention weight matrix at each step, it seems to be more time-consuming than single-step methods, so relevant experimental analysis also needs to be provided.
>
> **Response to Major Question 2:**
>
> Thank you for your comment. We will respond to the two questions above in order.
>
> ***Comparison with Single-step methods***
>
> We would like to clarify that the single-step observation methods may overlook crucial information and fail to capture the underlying motivations in our paper.
>
> Our reference to existing single-step methods at line 134 specifically relates to the approaches taken by Li (2024) and Wang (2023) in their calculation of saliency scores, which involve back propagating the loss from either the answer step or the final step.
>
> This methodology does not facilitate an understanding of phenomena such as the distraction caused by the presence of "160" in the demonstration as illustrated in Figure 2(b), which can lead LLMs to generate an incorrect token in the middle of a sentence as commented by reviewer Ngiq.
>
> Also, it is challenging to design a single-step intervention, as LLMs generate rationales prior to delivering a final answer in CoT reasoning tasks. Therefore, it seems unreasonable to intervene with the LLMs only at the last step, expecting them to produce a correct answer without addressing the already flawed rationales.
>
> Given the reasons outlined above, implementing single-step interventions proves challenging due to the absence of an analysis of the intermediate processes that would highlight the motivation for the intervention.
>
> On the contrary, in this paper, we first examine the distracting effects of few-shot CoT reasoning by dynamically analyzing the information flow between demonstration tokens and generated tokens at each step. Our proposed method, FAI, is based on the observation that tokens which have not undergone significant information aggregation in the CoT demonstrations are more likely to disrupt the generation of rationales. Consequently, FAI suppresses the impact of these tokens on the prediction at each step, preventing large language models (LLMs) from generating incorrect tokens influenced by these distractions.
>
> *In fact, our intervention method does not perform calculations at every step. Instead, it conducts a single overall calculation after one forward pass, making it highly efficient*. We will elaborate on this further below.
>
> ***Time-consuming issue***
>
> As discussed in the Major Question 1, the additional steps to run FAI using attention weights are as follows:
> 1. Input the CoT demonstrations into the model separately, run a single forward pass, and record the attention matrices for each head.
> 2. Based on the attention scores, identify the tokens in the demonstrations that did not exhibit significant aggregation, and record their corresponding layers and positions.
> 3. Input the CoT demonstrations and the question into the model. During the forward pass, set the attention scores to zero for the corresponding positions recorded in step 2 when generating each token.
>
> The operations in step 1 and step 2 need to be performed only once for a given problem. The additional computational cost introduced by step 3 on top of the existing generation is negligible. As a result, the computational cost of running FAI with the attention weights remains very low. *As highlighted by reviewer wLEA, the method does not bring extra computation gains (only a constant computational overhead)*.

---

> ### Author Response · Authors · 2024-11-22
> **Response to Reviewer k9ai (4/6)**
>
> **Major Question 3:**
>
> Questions about Figure 2.
>
> **Response to Major Question 3:**
>
> We sincerely apologize for any misunderstandings that may have occurred and would like to provide further clarification regarding the content of Figure 2.
>
> **Attention Saliency Pattern of Figure 2 (b) and (d)**
>
> *About information aggregation*: The selected tokens do not effectively gather information from other tokens in any of the preceding layers (i.e., the first 14 layers and the first 28 layers in Figure 2(b) and (d) respectively). The saliency from other tokens towards them shows similar patterns in earlier layers.
>
> *About impact on output token*: When the above unaggregated tokens exert a significant influence on the model's prediction at any layer, they are likely to disrupt the model's prediction of the subsequent token. Figure 2 illustrates this with layer 15 and 28; however, this phenomenon can also occur in any layer.
>
> **Attention Saliency Pattern of Figure 2 (a) and (c)**
>
> *About information aggregation*: The selected tokens can experience significant information aggregation from other tokens at any layer. We take layer 8 and 7 as examples to illustrate the information aggregation pattern in Figure 2(a) and (c) respectively (as Figure 2 has been revised, it is layer 5 and 3 in the previous version of manuscript).
>
> *About impact on output token*: Though the above aggregated tokens may demonstrate strong saliency towards the prediction position at any subsequent layer, they are less likely to distract the next token prediction of LLMs. This is illustrated in Figure 2 (a) and (c) with layer 29 and 30, respectively (layer 31 and 27 in the previous version of manuscript); however, similar phenomenon may also occur in other layers.
>
> This phenomenon highlights human cognitive tendencies to focus on prominent local elements while overlooking the broader context. When a token lacks substantial information aggregation, its hidden states retain significant isolated semantic information, making it more likely to become a focal point under certain conditions and influencing information processing dynamics.
>
> Notably, the FAI method we proposed is not subject to any layer or head, it adaptively identifies the tokens need intervened in every layer. Based on the layers and heads FAI identifies, the failure attention pattern can occur in any layer or head.
>
> **Major Question 4:**
>
> The experiments are not sufficient and should be verified on more types of models with different parameters.
>
> **Response to Major Question 4:**
>
> Thanks for the insightful advice. We further validate the effectiveness of FAI on GPT2-XL, GPT-NEO-2.7B and Llama-3-70B-Instruct as suggested.
>
> GPT-2 XL and GPT-NEO serve as two lower baselines, and the errors they produce on various evaluation benchmarks are primarily constrained by their inherent capabilities rather than being distracted by few-shot CoT prompting, particularly on datasets like Date Understanding and Last Letter. Nevertheless, applying FAI can still enhance the performance of these lower baselines. Specifically, utilizing FAI with GPT-2 XL and GPT-NEO leads to accuracy increases of 6.3% and 13.39%, respectively, on the AQuA dataset.
>
> Llama-3-70B-Instruct achieves the highest scores on various benchmarks, but applying FAI to this powerful baseline consistently leads to further improvements across almost all benchmarks, further demonstrating the effectiveness of FAI.
>
> |  method  | AQuA  | GSM8K | CSQA | Date | Sport | Last Letter
> |---|---|---|---|---|---|---|
> | GPT2-XL |  22.44 |   2.27  | 16.54 |  2.0 | 55.2 | 0.0 |
> | GPT2-XL + FAI | 28.74 |  2.88 | 16.63 | 2.0 | 55.2 | 0.0 |
> |GPT-NEO |  22.83 |  1.59  | 22.69 | 3.2 | 54.4 | 0.0 |
> |GPT-NEO + FAI |  36.22 | 2.50 |  23.26 | 3.6 | 55.2 | 0.0 |
> | Llama3-8B | 40.94 | 70.32 | 71.17 | 64.00 | 95.60 | 58.67 |
> | Llama3-8B + FAI | 46.85 | 71.24 | 74.28 |  65.60 | 96.00 | 62.00 |
> | Llama3-70B | 66.14 | 91.28 | 77.31 | 87.60 | 97.2 | 84.00 |
> | Llama3-70B +FAI | 66.53 | 91.28 | 78.62 | 88.0 | 98.0 | 85.33 |

---

> ### Author Response · Authors · 2024-11-22
> **Response to Reviewer k9ai (5/6)**
>
> **Major Question 5:**
>
> Reasons why samples with an accuracy rate greater than 90% as GSM_bad.
>
> **Response to Major Question 5:**
>
> To identify cases influenced by distractions for further examination, we select the 347 samples with an accuracy rate exceeding 90% and incorporate the demonstrations that resulted in incorrect answers, forming the GSM_bad set. This approach is based on the intuition that occasional mistakes are more likely to stem from distractions. Therefore, choosing a threshold of 90% is more representative than lower thresholds of 60% or 50%.
>
> Furthermore, we also conducted manual verification of the data to demonstrate that the samples are indeed influenced by the distracting effect.  We randomly sampled 180 of these samples for manual observation as stated in Appendix A.2.
>
> Naturally, We utilized the attention saliency metric from Figure 2 to pinpoint tokens in CoT demonstrations exhibiting the mentioned phenomenon and to investigate whether the error responses of the other categories are also caused by the reason. Based on this, it is estimated that about 60% of the erroneous responses in GSM_bad  are due to the distracting effect. The detailed process is described in Appendix A.2.
>
> Additionally, as suggested, we add the visualization of some cases in the revised manuscript which can be found in section A.5 of the Appendix.
>
> **Major Question 6:**
>
> Limitations of FAI.
>
> **Response to Major Question 6:**
>
> **What examples it tends to answer incorrectly that could otherwise be answered correctly without FAI?**
>
> As shown in Figure 4(a), following the application of FAI, the accuracy on GSM_good declined slightly from 100% to 96.58% while the accuracy on GSM_bad increases from 0% to 60.92%. This decrease can be attributed, in part, to some randomness that resulted in a few incorrect responses. Nonetheless, the overall performance of the FAI remains relatively robust.
>
> **What examples does it tend to answer incorrectly even with FAI?**
>
> FAI is capable of addressing issues stemming from token interference during demonstrations that result in incorrect answers. However, it does not offer substantial assistance for problems arising from the model's inherent limitations or other factors causing incorrect responses, such as the errors caused by repeated outputs analyzed in section A.2 of the Appendix.

---

> ### Author Response · Authors · 2024-11-22
> **Response to Reviewer k9ai (6/6)**
>
> **Minor Questions 1:**
>
> Why does the attention weight matrix in Figure 3 exclude the query question in Layer n but include it in Layer n+1?
>
> **Response to Minor Question 1:**
>
> We sincerely apologize for any confusion this may have caused, and we have revised Figure 3 to improve its clarity and understanding.
>
> Figure 3 illustrates how FAI identifies demonstration tokens that have not undergone significant information aggregation utilizing the attention weight matrix of demonstrations at Layer n. It then intervenes in the information transmission from these tokens to the prediction position in the attention weight matrix of Layer n+1, aiming to reduce the likelihood of the LLMs generating outputs distracted by these tokens.
>
> **Minor Question 2:**
>
> The description on lines 273 to 274 contradicts the description on lines 260 to 263.
>
> **Response to Minor Question 2:**
>
> We apologize for the typo. The description on lines 273 to 274 should be alpha larger than tau. Our experiments were conducted under the condition that alpha is greater than tau, but we mistakenly wrote otherwise. We appreciate you pointing out our error, and we have made the necessary corrections in the paper.
>
> **Minor Question 3:**
>
> There is no explanation of Table 3.
>
> **Response to Minor Question 3:**
>
> Thanks for the reminder and sorry for the confusions caused. We have added explanation of Table 3 in the revised manuscript.
>
> Table 3 presents the distribution of accuracy for each test sample across the 45 trials. As indicated in table 3, only 198 out of the 1319 samples received consistent responses—either always correct or always incorrect—across the various demonstrations. The remaining samples, which make up approximately 85% of the test set, yielded varied outcomes depending on the demonstration, suggesting that the potential for success with few-shot chain-of-thought reasoning is substantial, yet the risk of failure is equally significant.
>
> **Minor Question 4:**
>
> How is RAFR calculated?
>
> **Response to Minor Question 4:**
>
> RAFR (Rate of Answer Following the Rationale) is defined to measure whether model follows the pattern that generate rationales before giving the final answer. Below is a pair of typical examples:
>
> ***Question:***
>
> *Toulouse has twice as many sheep as Charleston. Charleston has 4 times as many sheep as Seattle. How many sheep do Toulouse, Charleston, and Seattle have together if Seattle has 20 sheep?*
>
> ***Solution that gives answer first***
>
> *160\nExplanation: Seattle has 20 sheep. Charleston has 4 times as many, so Charleston has 4 x 20 = 80 sheep. Toulouse has twice as many as Charleston, so Toulouse has 2 x 80 = 160 sheep. Together, they have 20 + 80 + 160 = 160 sheep.*
>
> ***Solution that answer follows the rationale***
>
> *Answer: Charleston has 4 times as many sheep as Seattle, so Charleston has 4*20 = 80 sheep. Toulouse has twice as many sheep as Charleston, so Toulouse has 2*80 = 160 sheep. Together, they have 20 + 80 + 160 = 260 sheep.*
>
>
> Initially, we design a prompt and using GPT-4 to identify whether the answer is given after the rationale. However we find that the format of Llama-3-8B-Instruct is quite fixed.  Therefore, for simplicity, we employ a rule based method to calculate RAFR (Rate of Answer Following the Rationale). Specifically, we detect whether there exists certain key words like “Explanation” in the beginning of the solution.
>
> **Minor Question 5:**
>
> What does “contrastive” in Figure 4 stand for?
>
> **Response to Minor Question 5:**
>
> The meaning of “contrastive” is explained in the Results and Analysis part of section 4.2.  In the contrastive setting, all information flow from demonstrations to prediction is blocked in each attention head. It serves as a contrastive experiment to further demonstrate the effectiveness of the proposed method.

---

> ### Author Response · Authors · 2024-11-24
> **Inquiry Regarding Rebuttal Feedback**
>
> Dear Reviewer k9ai,
>
> We sincerely appreciate your time and effort in reviewing our manuscript and offering valuable suggestions. We provided detailed clarifications in response to your questions a few days ago. If you have any additional feedback, concerns, or questions regarding our response, we would greatly appreciate hearing from you and welcome further discussion.

---

> > ### Comment · Reviewer_k9ai · 2024-11-25
> > **Response to rebuttal**
> >
> > I have carefully read the reviews from other reviewers and responses from the authors. The authors addressed my main concern about the technical novelty of the proposed method, I am convinced that there exists a distinction between single-step saliency scoring and its application to the CoT scenarios, which brings new insights into the understanding of the few-shot CoT inference. Also, I am happy to see that the proposed method does not bring extra computational workload. Thanks for the detailed responses from the authors, I raise my rating to $6$.

---

> ### Author Response · Authors · 2024-11-25
>
> Thanks again for your valuable comments, and we are so glad to hear that your concerns have been addressed.

---

### Official Review · Reviewer_wLEA · 2024-11-03

**Soundness:** 2
**Presentation:** 3
**Contribution:** 3
**Rating:** 6
**Confidence:** 4

**Summary:**

The authors focus on the failure mode of few-shot CoT. Specifically, they find that the demonstration selection will affect CoT's reasoning dynamic by distracting models with specific tokens. For example, models will use in-context examples, conditions, and numbers. To verify this phenomenon, authors first use a saliency map and shows that sometimes models put overwhelming attention on numbers of in-context examples, leading to an incorrect answer.

To solve the problem, authors use the self-attention score as an indicator of information aggregation and classify tokens that have high self-attention score as a distractor. Therefore authors remove these tokens when decoding to mitigate their effect.

Results on math reasoning, commonsense reasoning, and other tasks show the method's effectiveness in enhancing few-shot CoT performance.

**Strengths:**

* The topic is interesting and brings novel insight into understanding the few-shot CoT dynamic.

* The method is conceptually simple, implementation simple, and does not bring extra computation gains (only a constant computational overhead)

* The results are significant on various tasks with Llama 3 8B.

* Clear writing and figures.

**Weaknesses:**

* Authors use the aggregation of self-attention as an indicator of information flow. While it makes some sense intuitively, more supporting evidence is needed. For example, in the error case 2(b), can this method identify the same token as the saliency map?

* A case study is needed. Can authors provide several examples and highlight the tokens selected by the method? It will be very useful to observe the approach's token selection to understand its advantages and drawbacks.

* More analysis of the saliency map. Figure 2 shows the failure attention pattern on specific layers and heads, but it is unclear if this only occurs in a few layers and heads. What do other heads' and layers' attention patterns look like?

**Questions:**

* Can authors add zero-shot results in Table 2 and Table 4? It will be helpful to know how better the few-shot CoT is.

* Does the attention-dropping only apply to the tokens between decoding and in-context examples? For example, will tokens between different in-context examples also have attention-dropping, and will different decoding tokens have attention-dropping among themselves?

* line 73: "has not experienced significant information aggregation" should be the case that alpha is larger than tau rather than smaller than tau?

* line 45, 133, 139: \citet -> \citep

---

> ### Author Response · Authors · 2024-11-21
> **Response to Reviewer wLEA (1/2)**
>
> We sincerely appreciate your valuable feedback and positive evaluations of our paper. We have thoughtfully considered your insightful suggestions and will address your concerns as outlined below.
>
> **Weakness 1:**
>
>  Authors use the aggregation of self-attention as an indicator of information flow. While it makes some sense intuitively, more supporting evidence is needed. For example, in the error case 2(b), can this method identify the same token as the saliency map?
>
> **Response to Weakness 1:**
>
> Thanks for the insightful comment. We have included a comparison of analyses based on attention scores and saliency scores in Figure 6 and Figure 7 of section A.4 in the Appendix. These figures demonstrate that the behavior of attention scores closely resembles that of saliency scores across various cases. This approximation can be partially reflected in Equation 1, as the saliency score is defined as the Hadamard product of the attention score and the corresponding gradient.
>
> Therefore, attention scores can serve as an approximation of saliency scores. Additionally, Figure 8 in Section A.5 of the appendix visualizes the tokens selected by our proposed method in the error case 2(b), confirming that the same tokens can indeed be identified.
>
>
>
> **Weakness 2:**
>
> A case study is needed. Can authors provide several examples and highlight the tokens selected by the method? It will be very useful to observe the approach's token selection to understand its advantages and drawbacks.
>
> **Response to Weakness 2:**
>
> Thanks for the suggestion. We have provided an analysis of tokens identified by FAI in the Table 6 of the original manuscript which shows that many of the most frequest tokens selected are mathematical symbols or numbers,   such as “<<”, “=”.
>
> Additionally, we have included more case analyses illustrating the tokens selected as suggested. Figures 8, 9, and 10 in Section A.5 of the appendix display the tokens identified by the FAI along with the corresponding saliency visualizations. These figures highlight that the tokens identified by FAI encompass those that resulted in incorrect model responses, as revealed through the saliency visualization analysis. This further underscores the effectiveness of the proposed FAI method.
>
> It is worth noting that the tokens that caused the model to make mistakes in these case analyses are also largely consistent with the analysis in Table 6.
>
> **Weakness 3:**
>
> More analysis of the saliency map. Figure 2 shows the failure attention pattern on specific layers and heads, but it is unclear if this only occurs in a few layers and heads. What do other heads' and layers' attention patterns look like?
>
> **Response to Weakness 3:**
>
> We apologize for any confusion caused by Figure 2. We would like to further clarify the meaning of Figure 2.
>
> **Attention Saliency Pattern of Figure 2 (b) and (d)**
>
> _*About information aggregation:*_ The selected tokens do not effectively gather information from other tokens in any of the preceding layers (i.e., the first 14 layers and the first 28 layers in Figure 2(b) and (d) respectively). The saliency from other tokens towards them shows similar patterns in earlier layers.
>
> _*About impact on output token:*_ When the above unaggregated tokens exert a significant influence on the model's prediction at any layer, they are likely to disrupt the model's prediction of the subsequent token. Figure 2 illustrates this with layer 15 and 28; however, this phenomenon can also occur in any layer.
>
> **Attention Saliency Pattern of Figure 2 (a) and (c)**
>
> _*About information aggregation:*_ The selected tokens can experience significant information aggregation from other tokens at any layer. We take layer 8 and 7 as examples to illustrate the information aggregation pattern in Figure 2(a) and (c)  respectively (as Figure 2 has been revised, it is layer 5 and 3 in the previous version of manuscript).
>
> _*About impact on output token:*_ Though the above aggregated tokens may demonstrate strong saliency towards the prediction position at any subsequent layer, they are less likely to distract the next token prediction of LLMs. This is illustrated in Figure 2 (a) and (c) with layer 29 and 30, respectively (layer 31 and 27 in the previous version of manuscript); however, similar phenomenon may also occur in other layers.
>
> This phenomenon highlights human cognitive tendencies to focus on prominent local elements while overlooking the broader context. When a token lacks substantial information aggregation, its hidden states retain significant isolated semantic information, making it more likely to become a focal point under certain conditions and influencing information processing dynamics.
>
> Notably, the FAI method we proposed is not subject to any layer or head, it adaptively identifies the tokens need intervened in every layer. Based on the layers and heads FAI identifies, the failure attention pattern can occur in any layer or head.

---

> ### Author Response · Authors · 2024-11-21
> **Response to Reviewer wLEA (2/2)**
>
> **Question 1**
>
> Can authors add zero-shot results? It will be helpful to know how better the few-shot CoT is.
>
> **Response to Question 1:**
>
> Thanks for the valuable suggestion. We have supplemented the zero-shot results as suggested.
>
> The results demonstrate that few-shot CoT usually outperforms zero-shot approaches. In particular tasks, such as Sport Understanding and Last Letter Concatenation, few-shot CoT serves as a description of the task; without it, the models are unable to produce correct answers.
>
> FAI is able to further enhance the performance based on few-shot CoT, highlighting its effectiveness.
>
> |  method  | AQuA  | GSM8K | CSQA | Date | Sport | Last Letter |
> |---|---|---|---|---|---|---|
> | Llama3-8B Zeroshot| 16.14 | 21.60 | 0.82 | 8.0 | 0 | 0 |
> | Llama3-8B | 40.94 | 70.32 | 71.17 | 64.00 | 95.60 | 58.67 |
> | Llama3-8B + FAI | 46.85 | 71.24 | 74.28 |  65.60 | 96.00 | 62.00 |
>
> |  number of shots  | 1-shot  |  | 2-shot | | 4-shot | |  6-shot | |
> |---|---|---|---|---|---|---|---|---|
> |  methods | Retrieval  | Random | Retrieval  | Random | Retrieval  | Random | Retrieval  | Random
> | Llama2-13B Zeroshot| 29.57 | | | |  |  |  |  |
> | Llama2-13B | 30.55 | 32.45  | 33.59 | 32.52 | 33.51 | 33.97 |  34.65 | 34.04 |
> | Llama2-13B + FAI | 34.34 | 32.45 | 33.66 |  34.27 | 34.80 | 35.86 | 35.18 | 36.77 |
> | Llama3-8B Zeroshot| 21.60 | | | |  |  |  |  |
> | Llama3-8B | 67.78 |  69.29  | 68.99 | 73.62 | 71.65 | 73.09 |   68.84 |  71.65 |
> | Llama3-8B + FAI |  67.78 | 69.90 | 71.27 |  73.77 | 73.54 | 74.30 | 71.95 |  75.21 |
> | Mistral-7B Zeroshot| 33.89 | | | |  |  |  |  |
> | Mistral-7B | 35.33 |  35.86  | 36.24 | 38.06 | 38.13 | 39.73 |   36.62 |  37.30 |
> | Mistral-7B + FAI |  36.09 | 37.15 | 39.27 |  38.59 | 41.93 | 41.55 | 38.89 |  38.29 |
>
> **Question 2**
>
> Does the attention-dropping only apply to the tokens between decoding and in-context examples? For example, will tokens between different in-context examples also have attention-dropping, and will different decoding tokens have attention-dropping among themselves?
>
> **Response to Question 2:**
>
> Thanks for the insightful comment.  Currently, the attention-dropping only apply to the tokens between decoding and in-context examples.
>
> The necessity of leveraging factual information from previously generated tokens during the decoding process is not inherently detrimental. Therefore, applying attention-dropping across different decoding tokens may poses risks. The question of whether attention-dropping should be employed between distinct in-context examples is intriguing and merits further exploration, which we plan to address in future research.
>
> Notably, the experimental results in Table 4 demonstrate that the proposed method is able to gain further improvement regardless of the number of CoT demonstrations.
>
> **Question 3:**
>
> line 73: "has not experienced significant information aggregation" should be the case that alpha is larger than tau rather than smaller than tau?
>
> **Response to Question 3:**
>
> We apologize for the typo, and you are correct that alpha should be larger than tau. Our experiments were conducted under the condition that alpha is greater than tau, but we mistakenly wrote otherwise. We appreciate you pointing out our error, and we have made the necessary corrections in the paper.
>
> **Question 4:**
>
> line 45, 133, 139: \citet -> \citep
>
> **Response to Question 4:**
>
> We apologize for the errors we made and have thoroughly reviewed the entire paper, correcting all citations.

---

> ### Author Response · Authors · 2024-11-24
> **Inquiry Regarding Rebuttal Feedback**
>
> Dear Reviewer wLEA,
>
> We sincerely appreciate your time and effort in reviewing our manuscript and offering valuable suggestions. We provided detailed clarifications in response to your questions a few days ago. If you have any additional feedback, concerns, or questions regarding our response, we would greatly appreciate hearing from you and welcome further discussion.

---

> > ### Comment · Reviewer_wLEA · 2024-11-25
> >
> > Thanks for the authors' rebuttal. I've fully read it.
> >
> > The authors address my concerns on qualitative analysis, making me better understand the paper.
> >
> > I keep my tendency to accept. Considering the authors' claims need thorough qualitative examples to justify (i.e., the attention intervention functionality), I think 6 is appropriate based on the current manuscripts.

---

> ### Author Response · Authors · 2024-11-25
>
> Thanks again for your valuable comments, and we are so glad to hear that your concerns have been addressed.

---

### Official Review · Reviewer_Ngiq · 2024-11-05

**Soundness:** 2
**Presentation:** 2
**Contribution:** 3
**Rating:** 6
**Confidence:** 4

**Summary:**

This paper analyzes how tokens in demonstration examples influence chain-of-thought reasoning. Based on their findings, the authors propose a method to block token information flow using attention scores. Experimental results demonstrate the effectiveness of this approach.

**Strengths:**

- The analysis results are interesting and insightful.

**Weaknesses:**

- The analysis is done with saliency scores while the proposed method is based on attention scores. It is not clear to me that if attention scores will exhibit similar behavior to saliency scores. If the proposed method uses attention scores, the analysis should also be presented using attention scores to ensure consistency.
- Section 3.3 needs further elaboration. In Figure 1, it seems that certain tokens in the demonstration examples receive "high scores," which could potentially disrupt the reasoning process. My understanding is that the authors aim to adjust attention to reduce this distraction. However, in intervening with the information flow (Section 3.3), the authors block attention only for tokens with "no significant information aggregation" rather than focusing on high-impact tokens. This seems like a mismatch. Please correct me if I misunderstand.
- Figure 1 is somewhat misleading. Figure (a) and (c) show predictions for the next token at the start of a new sentence, while (b) and (d) show predictions in the middle of a sentence or even during equation completion. This difference may explain the substantial variation in score distribution. I assume that the cases in (a) and (c) are simpler and may not effectively support the authors' claims. I would like to see some other examples for (a) and (c), particularly those involving equation completion, to demonstrate the motivation.

**Questions:**

Please see above.

---

> ### Author Response · Authors · 2024-11-21
> **Response to Reviewer Ngiq**
>
> We sincerely appreciate your valuable insights and constructive feedback. We have diligently completed all the case studies and analyses you recommended. The revised manuscript now includes a comparison of analyses based on attention scores and saliency scores, as well as additional cases in Figure 2 that more effectively support our claims. Our detailed responses are as follows:
>
> **Weakness 1:**
>
> It is not clear to me that if attention scores will exhibit similar behavior to saliency scores.
>
> **Response to Weakness 1:**
>
> Thanks for the insightful comment. We have included a comparison of analyses based on attention scores and saliency scores in Figure 6 and Figure 7 of section A.4 in the Appendix. These figures demonstrate that the behavior of attention scores closely resembles that of saliency scores across various cases. This approximation can be partially reflected in Equation 1, as the saliency score is defined as the Hadamard product of the attention score and the corresponding gradient.
>
> Therefore, attention scores can serve as an approximation of saliency scores. Furthermore, attention scores are significantly more computationally efficient than saliency scores, and their normalized nature enhances adaptability in measuring the degree of information aggregation for tokens.
>
> Considering these factors, our proposed method utilizes attention scores. We have added detailed clarifications in the paper to avoid any potential confusion.
>
> **Weakness 2:**
>
> Section 3.3 needs further elaboration.
>
> **Response to Weakness 2:**
>
> I believe you meant to refer to Figure 2.  We sincerely apologize for any misunderstandings that may have arisen and would like to take this opportunity to clarify the meaning of Figure 2.
>
> In Figure 2, certain tokens (e.g., token 160 in Figure 2 (b)) in the demonstration consistently exhibit a *high self-saliency score*. This indicates that there is *no significant information aggregation* from preceding tokens to these particular tokens. These tokens sometimes have a *high impact* on the next predicted token of LLMs and could potentially disrupt the reasoning process.
>
> To mitigate the potential distractions, we implement attention blocking for tokens with *high self-attention scores* (which signal a lack of significant information aggregation, or means high-impact tokens) as discussed in section 3.3.  We have revised this section for clarity and comprehension.
>
> **Weakness 3:**
>
> The cases in (a) and (c) are simpler and may not effectively support the authors' claims. I would like to see some other examples for (a) and (c), particularly those involving equation completion, to demonstrate the motivation.
>
> **Response to Weakness 3:**
>
> I believe you were referring to Figure 2. Your suggestions are immensely valuable. The bias introduced by the varying positions of output tokens needs to be considered for removal.
>
> To address this issue, we have followed your recommendation and replaced (a) and (c) in Figure 2 with instances where specific tokens significantly influence model predictions during these calculations. The new cases demonstrate that tokens, which undergo substantial information aggregation from others, are less likely to distract LLMs, even when they possess a high saliency score during equation completion. The updated Figure 2 is included in the revised manuscript we have just uploaded.

---

> > ### Comment · Reviewer_Ngiq · 2024-11-22
> >
> > Thanks for the response. It addressed my concern about weakness 2 and 3.
> >
> > However, After check Figure 6 and 7, I found that the behavior of attention scores and saliency scores can be different. For example, in Figure 6(b), (btw, you miss (a)(b)(c)(d) in Figure 6), the highest attention score is not `160` but the space after `6 *`. So the figure should show the attention score of the space rather than `160` to check if it has a high self-attention score. Similarly, I can see that the highest attention scores are different from highest saliency scores in Figure 6(d), 7(b), and 7(d).
> >
> > Given that, I think the analysis related to Table 1 and Appendix A.2 need to be retaken for attention scores as well, as they were done based on saliency scores.

---

> ### Author Response · Authors · 2024-11-23
> **Response to Reviewer Ngiq**
>
> Thank you for your prompt response and we are delighted to provide the following response for your remaining question.
>
> ## The highest attention score is not 160 but the space after 6 * in Figure 6 (b)
>
> We would like to further clarify the similarity in performance between saliency scores and attention scores from two perspectives.
>
> **Information aggregation:**
>
> As demonstrated in Figures 6 and 7, the behavior of attention scores closely parallels that of saliency scores in indicating whether a token has experienced significant information aggregation. Tokens characterized by a high self-saliency score also tend to exhibit a correspondingly high self-attention score, whereas tokens with a low self-saliency score typically display a similarly low self-attention score.
>
> **Impact on output token:**
>
> Your observations are very careful and precise. The space after "6 *" has the highest attention score on the prediction in Figure 6 (b), however, "160" still has a very high attention score.
>
> In fact, you may notice that "160" (i.e., the token with the highest saliency score) holds the highest attention score among the tokens _in the demonstration_, which is also the case in Figure 6(d), 7(b), and 7(d), while the space after "6 *" and other tokens with higher attention scores are in the generated rationale or in the question.
>
> Our paper specifically explores the impact of _tokens in few-shot CoT demonstrations_ on the _next token to be predicted by the model_. The method we propose aims to identify and block the distractive flow of information between these two, rather than the attention scores between the forthcoming token and those already produced.
>
> Therefore, the phenomenon shown in Figure 6 and Figure 7 about the saliency scores and attention scores of tokens in the demonstrations on the subsequent token to be generated by model is largely consistent for our method.
>
> As demonstrated in [1], when generating next token, models trained with next-token prediction would naturally tend to assign a high attention score on the last generated token. Consequently, the high attention score on the space after "6 *" may not be detrimental.
>
> Nonetheless, the patterns of attention scores of tokens in question or generated rationale are intriguing and merit further exploration, which we plan to address in future research.
>
> [1] Li et al. Mechanics of Next Token Prediction with Self-Attention
>
> ## The analysis related to Table 1 and Appendix A.2 need to be retaken for attention scores as well, as they were done based on saliency scores.
>
> First, we would like to clarify the objectives of analysis in A.2. Then, following your valuable suggestions, we have re-conducted the analysis in A.2 using attention scores.
>
> **The purpose of conducting the analysis in A.2:**
>
> The classification of error cases in Table 1 is derived from manual observation and is unrelated to the use of saliency scores or attention scores. In Appendix A.2, we conducted an analysis of the saliency scores for these cases, which _confirmed that a significant portion of them was indeed influenced by tokens in the demonstrations that led to incorrect answers_.
>
> Our experimental results presented in Section 4.2 and Figure 4(a) demonstrate that, by employing our proposed FAI method with attention scores, we were able to increase the accuracy of answers on these cases (i.e., GSM_bad) from 0% to 60.92%, further validating that attention scores can serve as an alternative to saliency scores.
>
> **Retake analysis based on attention scores**
>
> As a reminder, in Section A.2, we randomly sampled 10 cases for each of the four error types to conduct a more in-depth analysis based on saliency scores. We found that the phenomena observed in 10 out of 10 IF samples, 9 out of 10 MC samples, and 8 out of 10 RS samples were consistent with those shown in Figure 2, whereas it was challenging to observe this phenomenon in the RO samples.
>
> We re-analyzed these samples using attention scores, and our conclusions were largely consistent with those derived from saliency scores. In the 10/10 IF samples and 9/10 MC samples, as well as in 6/10 RS samples, we observed the same phenomenon: specific tokens in the demonstration did not significantly accumulate attention scores but were assigned high attention scores for the output token, which led to the model producing incorrect tokens.
>
> Thank you once again for valuable feedback.

---

> > ### Comment · Reviewer_Ngiq · 2024-11-24
> >
> > Thanks for the clarification. I've increased my score.

---

> > > ### Author Response · Authors · 2024-11-24
> > >
> > > Thanks a lot for your valuable reply, we appreciate your effort and time, and we are so glad to hear that your concerns have been addressed.

---

### Meta-Review · Area_Chair_r1wt · 2024-12-20

**Metareview:**

This paper investigates the Few-shot Chain-of-Thought dynamics in LLMs by identifying and mitigating the distracting influence of specific tokens on the model's reasoning process. The proposed Few-shot Attention Intervention method dynamically adjusts attention weights to suppress the effect of such tokens, resulting in significant performance improvements on multiple benchmarks.

The reviewers raised some questions about the consistency between the use of saliency scores for analysis and attention scores for intervention, the clarity of the methodology, and the representation of examples that might not fully support the claims. During the rebuttal, the authors addressed these issues by clarifying the methodologies and providing additional examples and explanations, which generally satisfied the reviewers.

**Additional Comments On Reviewer Discussion:**

Nil

---

### Decision · Program_Chairs · 2025-01-22

Accept (Poster)